# Broad substrate scope C-C oxidation in cyclodipeptides catalysed by a flavin-dependent filament

Emmajay Sutherland [1,2], Christopher J. Harding [1], Tancrède du Monceau de Bergendal [1], Gordon J. Florence[3], Katrin Ackermann[3], Bela E. Bode [3], Silvia Synowsky[4], Ramasubramanian Sundaramoorthy[5] & Clarissa Melo Czekster [1]

Cyclic dipeptides are produced by organisms across all domains of life, with many exhibiting anticancer and antimicrobial properties. Oxidations are often key to their biological activities, particularly C-C bond oxidation catalysed by tailoring enzymes including cyclodipeptide oxidases. These flavin-dependent enzymes are underexplored due to their intricate three-dimensional arrangement involving multiple copies of two distinct small subunits, and mechanistic details underlying substrate selection and catalysis are lacking. Here, we determined the structure and mechanism of the cyclodipeptide oxidase from the halophile *Nocardiopsis dassonvillei* (NdasCDO), a component of the biosynthetic pathway for nocazine natural products. We demonstrated that NdasCDO forms filaments in solution, with a covalently bound flavin mononucleotide (FMN) cofactor at the interface between three distinct subunits. The enzyme exhibits promiscuity, processing various cyclic dipeptides as substrates in a distributive manner. The reaction is optimal at high pH and involves the formation of a radical intermediate. Pre-steady-state kinetics, a significant solvent kinetic isotope effect, and the absence of viscosity effects suggested that a step linked to FMN regeneration controlled the reaction rate. Our work elucidates the complex mechanistic and structural characteristics of this dehydrogenation reaction, positioning NdasCDO as a promising biocatalyst and expanding the FMN-dependent oxidase family to include enzyme filaments.

Secondary metabolites of microbial origin are a vital but largely unexplored source of compounds with untapped therapeutic potential. Across all domains of life, cyclic dipeptides (CDPs) exist either independently or embedded as part of larger, more complex natural products[1]. Distinguished by their 2,5-diketopiperazine core, CDPs are considered privileged frameworks that resist proteolysis, exhibit easy gut absorption, and can permeate the blood-brain barrier[2]. These properties have since facilitated the use of CDPs as antibacterial[3],

[1]University of St Andrews, School of Biology, North Haugh, Biomolecular Sciences Building, St Andrews, UK. [2]University of Washington, Department of Chemistry, Seattle, WA, USA. [3]University of St Andrews, EaStCHEM School of Chemistry, North Haugh, Purdie Building, St Andrews, UK. [4]University of St Andrews, BSRC Mass Spectrometry and Proteomics Facility, North Haugh, Biomolecular Sciences Building, St Andrews, UK. [5]Laboratory of Chromatin Structure and Function, MCDB, School of Life Sciences, University of Dundee, Dundee, UK. ✉e-mail: R.Z.Sundaramoorthy@dundee.ac.uk; cmc27@st-andrews.ac.uk

antifungal[4], antiviral[5] and anticancer[2,6] agents. Whilst the precise functions of these molecules remain elusive, emerging evidence suggests their involvement in quorum sensing and transkingdom signalling[7,8].

Presently, three documented biosynthetic routes lead to a variety of cyclic dipeptides: nonribosomal peptide synthetases (NRPSs)[9], cyclodipeptide synthases (CDPSs)[10], and the recently discovered arginine-containing cyclodipeptide synthases (RCDPSs)[11]. Biosynthetic gene clusters for CDPSs have been predominantly identified in bacteria, whereas NRPSs and RCDPSs from eukaryotes have been described. NRPSs, with their large multi-modular architecture, are adept at utilising inactivated amino acids to generate cyclic dipeptide products[12]. In contrast, both CDPSs and RCDPSs utilize aminoacylated-tRNAs as substrates, therefore competing with protein translational processes[13]. Moreover, biosynthetic gene clusters encoding CDPSs often harbour tailoring enzymes that can further modify the CDP scaffold, commonly through methylation, prenylation and oxidation[14]. Specifically, oxidation can be catalysed by cytochrome P450s[15] and cyclodipeptide oxidases[16] (CDOs). Supplementary Fig. 1 depicts the genomic context of NdasCDO and similar biosynthetic gene clusters according to webflags[17].

CDOs enable the facile incorporation of a Cα-Cβ double bond into the 2,5-diketopiperazine backbone (Fig. 1). Despite their discovery in 2001[18], not much mechanistic insight is available on the reaction catalysed by CDOs. Initial investigations demonstrated that this enzyme family is composed of two subunits – A and B – where subunit A (22 kDa) is approximately twice the size of subunit B. These, together with a covalently bound flavin cofactor, comprise the catalytically competent CDO complex[19]. Within gene clusters containing CDOs, the DNA sequences encoding subunits A and B overlap by 20–30 nucleotides, a feature postulated to act as a natural regulatory mechanism governing the expression of both subunits[19]. Moreover, all attempts to express each subunit individually yielded inactive forms, emphasising that complex formation is key to enzyme functionality[20]. While it is established that CDOs employ molecular oxygen for dehydrogenation, producing hydrogen peroxide as a by-product, the overarching catalytic mechanism remains largely undetermined. Recently, it was determined that a member of the CDO family formed a heterotetrameric filamentous structure from the two individual subunit dimers[21]. Enzymatic activity was highly dependent on filament formation which necessitated the simultaneous expression of both subunits; attempts at individual subunit purification were also unsuccessful by Giessen et al. While enzyme filaments have been previously described[22], structural and mechanistic characterisations of these proteins are difficult, thus limiting a comprehensive understanding of reaction mechanism and structural data.

Notably, CDOs feature in four documented biosynthetic pathways: albonoursin from *Streptomyces noursei*[19]; nocazine from *Nocardiopsis dassonvillei*[16]; guanitrypmycin from *Streptomyces monomycini*[23]; and purincyclamide from *Streptomyces chrestomyceticus*[24]. Moreover, oxidized CDPs are building blocks for the natural product pulcherrimin, which acts as an iron chelator[25], and phenylahistin, a precursor to the anticancer drug Plinabulin[26]. Synthetic routes for generating specific oxidized CDPs are scarce, and yield side products that require lengthy purification steps. Biocatalysis utilizing CDOs has previously involved in vivo systems, wherein media supernatants were screened for potential oxidized cyclodipeptide products[27]. These efforts have successfully combined CDPSs and CDOs from different organisms within a singular vector, yielding a diverse array of dehydrogenated CDPs[20]. Nevertheless, a deeper molecular understanding of CDO structure and mechanism promises to enhance their utility as biocatalysts. By exploring reaction conditions to achieve optimal enzymatic efficiency whilst maintaining CDO stability, these enzymes can be harnessed as powerful tools for facilitating the introduction of C-C double bonds into cyclodipeptides.

The substrate scope of CDOs is varied with a singular CDO capable of accepting numerous CDPs. However, the reaction maintains exquisite selectivity for Cα-Cβ oxidation where FMN regeneration does not require additional cofactors. In this work we describe an in-depth characterisation of NdasCDO, the cyclodipeptide oxidase found in the nocazine biosynthetic pathway of *Nocardiopsis dassonvillei*[16]. Combining a plethora of biochemical and biophysical techniques including cryo-EM, molecular docking, electron paramagnetic resonance (EPR) spectroscopy, and kinetic assays in steady and pre-steady states, we determined substrate scope and rate limiting steps of the reaction, unveiling crucial features of

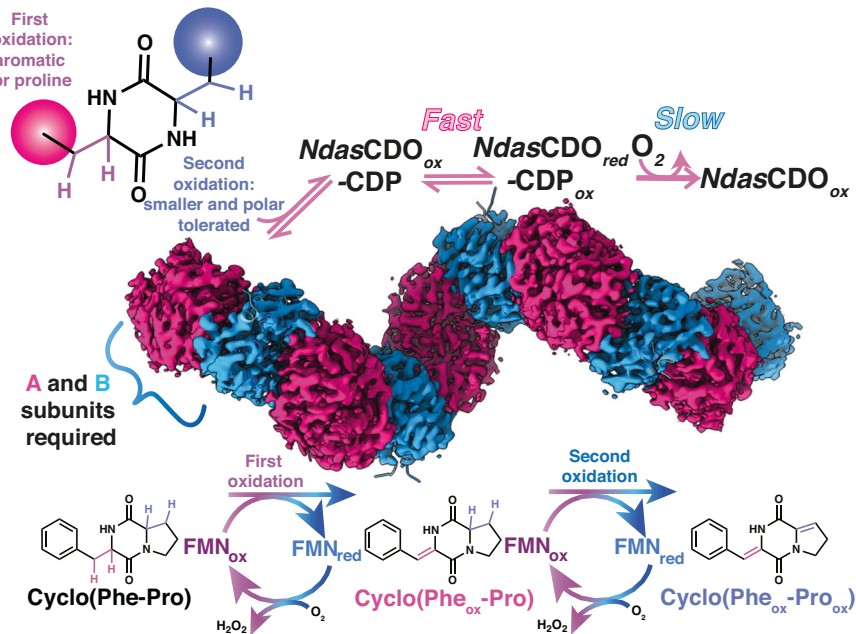

**Fig. 1 | Overview of NdasCDO-catalysed reaction.** Top: CDP substrate can contain one (if cyclic dipeptide contains a glycine) or two amino acid side chains which can undergo oxidation. Binding of CDP substrate is followed by fast peptide oxidation then by slow oxidation of the FMN cofactor to regenerate catalytic competent enzyme. Both A and B subunits are required for activity. Bottom: Illustration of two consecutive oxidation events which can take place with some CDP substrates.

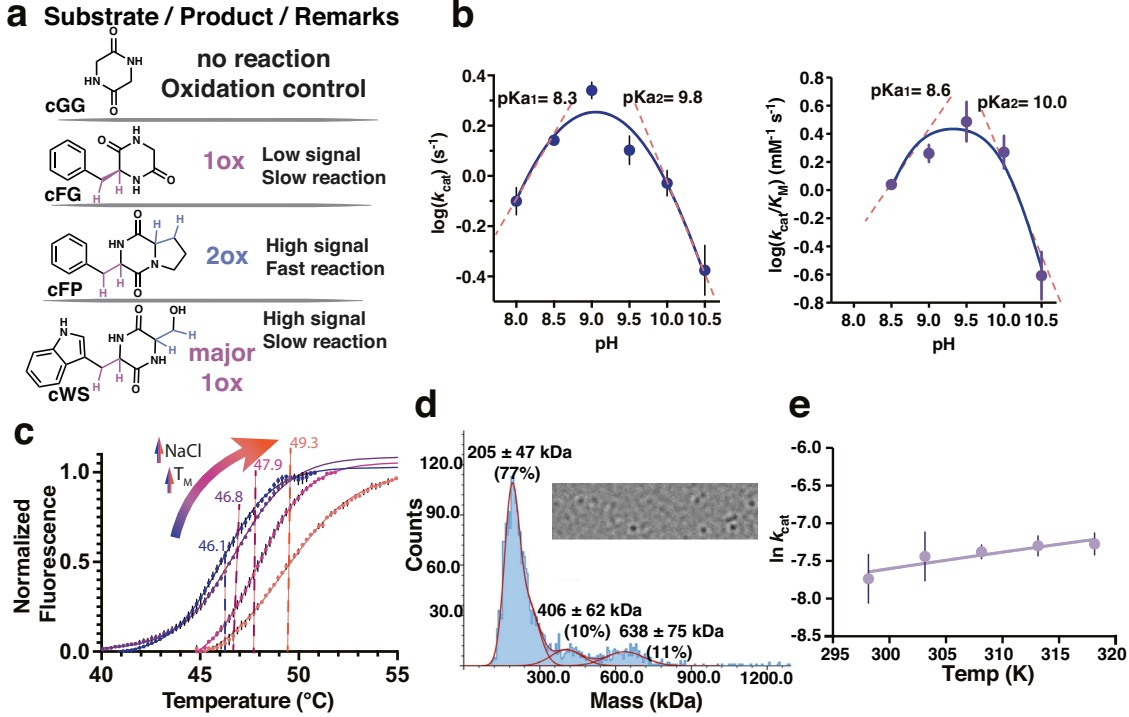

**Fig. 2 | Substrates used to probe mechanism of NdasCDO, effect of pH, temperature and salt concentration in the reaction. a** Criteria for usage of distinct CDP substrates for different experiments: cGG is a reaction control as it cannot be oxidized; catalysis with cFG is slow and a single oxidation is possible – oxidation leads to a small change in absorbance at 297 nm; cFP undergoes fast oxidation, and two oxidations are possible – oxidation leads to a large change in absorbance at 312 nm; (hydrogen atoms depicted purple when participating in the first oxidation and blue when participating in the second oxidation rection); cWS undergoes slow oxidation, and although two oxidations are possible, second is less favourable – oxidation leads to a large change in absorbance at 297 nm; **b** pH dependence of log $(k_{cat})$ and log $(k_{cat}/K_M)$ for varied cFG (blue and purple, respectively) examined using a mixed buffer system. Data were obtained at 30 °C whilst varying the concentration of substrate. log $(k_{cat})$ data are fit to a 1-proton dependence (Eq. (2)), log

$(k_{cat}/K_M)$ are fit to a 2-proton dependence equation (Eq. (3)), shown as mean values ± standard error of the fit across three individual replicates. Errors for $k_{cat}/K_M$ and $k_{cat}/K_M$ were propagated appropriately. **c** Melting curves using differential scanning fluorimetry (DSF), raw data (circles) fitted to a Boltzmann sigmoidal equation (lines); $T_m$ values after fitting are shown with varying NaCl concentrations. **d** Top, snapshot of mass photometry movie, which was recorded for 120 s. Data was processed using the DiscoverMP software to generate the mass distribution histograms displayed in the bottom. Under these conditions a predominant species of 205 ± 47 kDa is observed. **e** Temperature dependence of $k_{cat}$, data were fitted to Eq. (1) yielding $\Delta H^{\ddagger} = 4.06 \pm 2.42$ kcal mol$^{-1}$ and $\Delta S^{\ddagger}$ of $-0.049 \pm 0.008$ kcal mol$^{-1}$. Data shown as mean values ± standard error of the fit across three individual replicates.

NdasCDO as an FMN-dependent filament enzyme catalysing CDP dehydrogenation.

## Results and discussion
### Covalent FMN cofactor
NdasCDO purified with a yellow colour and characteristic absorption spectra for flavin-dependent enzymes (Supplementary Fig. 4a). Protein intact mass spectrometry showed a mass shift of +907 Da for the A subunit dimer (Supplementary Fig. 4b–d), indicative of one FMN cofactor bound per subunit. Trypsin digestion and peptide fingerprinting confirmed the assignment of Cys121 as the covalent attachment site for FMN. Prior work with CDOs under anaerobic conditions observed no cyclic dipeptide oxidation[18], and therefore the catalytic cycle starts with the cofactor as the oxidized quinone form of the cofactor (quinone; $Fl_{ox}$). Covalently attached flavin cofactors are observed in approximately 10% of flavoenzymes, and this attachment has been proposed to modulate cofactor redox potential, therefore facilitating catalysis of thermodynamically challenging reactions[28,29]. Additionally, the reaction catalysed by NdasCDO bears similarity to internal monooxygenases, as the same substrate is oxygenated (or dehydrated in the case of NdasCDO) by the enzyme and also subsequently acts as electron donor in flavin reduction (Fig. 2a)[30].

*Nocardiopsis dassonvillei* is a halophilic bacterium, with optimal growth conditions containing up to 3.4 M NaCl[31], therefore we explored whether high salt conditions were required for protein

stability and activity. Folding and melting temperatures of NdasCDO revealed a Tm of 46 °C with 0.2 M NaCl, and that higher salt concentrations were associated with higher melting temperatures (Fig. 2c, also discussed further below under "Oligomeric state"). Next, we analysed the temperature dependence of cFP oxidation catalysed by NdasCDO (individual Michaelis–Menten curves available in Supplementary Fig. 3a). Figure 2e depicts a slight curvature in the temperature range under evaluation. However, we opted for a simple model using a linear fit to the Eyring Equation without a contribution from activation heat capacity, further discussed below, with fitted values for the temperature dependence of $k_{cat}$ of $\Delta H^{\ddagger} = 4.06 \pm 2.42$ kcal mol$^{-1}$ and $\Delta S^{\ddagger}$ of $-0.049 \pm 0.008$ kcal mol$^{-1}$ (Supplementary Fig. 4b). In our standard reaction conditions (30 °C), a $\Delta G^{\ddagger} = 16.5$ kcal mol$^{-1}$ would be observed, mostly driven by entropic contributions. The relevance of entropic contributions in enzyme catalysis has been discussed in the context of the importance of interactions between solvent and outer protein surface[32], and whether the large complex formed by subunits A and B plays a role remains to be determined.

### pH dependence
NdasCDO depicted bell shaped profiles for both $k_{cat}$ and $k_{cat}/K_M$ (Fig. 2b). Best fitted values were obtained with a single ionizable group on each side of the $k_{cat}$ profile (p$K_{a1} = 8.3 \pm 0.1$, p$K_{a2} = 9.8 \pm 0.1$), but with two ionizable groups on each side of the $k_{cat}/K_{M\text{-cFP}}$ profile (p$K_{a1} = 8.6 \pm 0.1$, p$K_{a2} = 10.0 \pm 0.1$). These experiments were performed

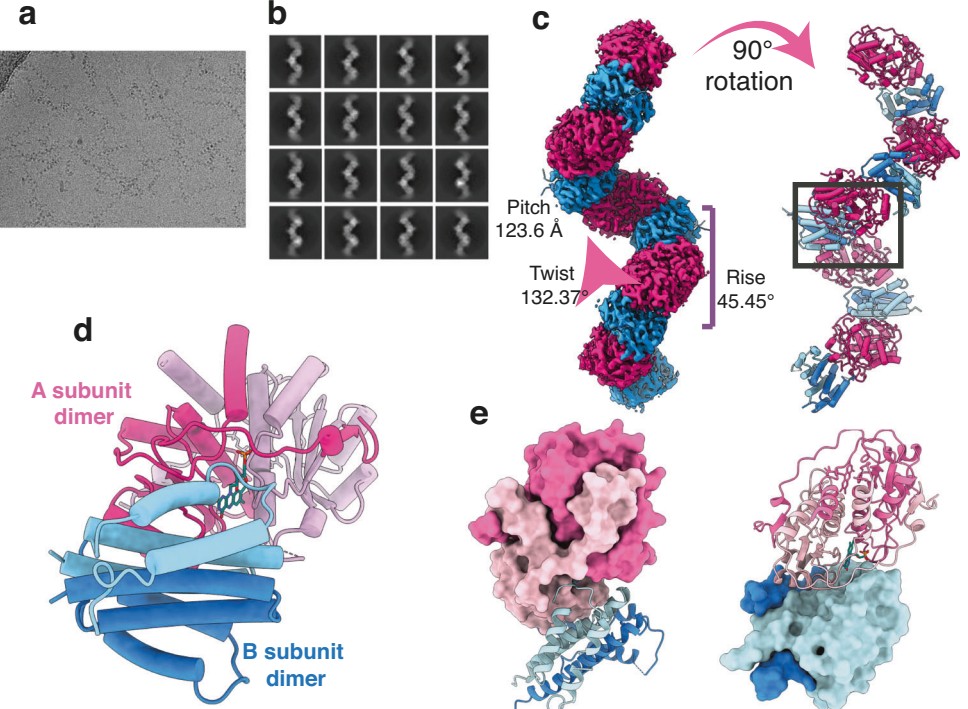

**Fig. 3 | NdasCDO filament structure revealed by cryo-EM.** For all panels, A subunits are shown in shades of pink and B subunits in shade of blue. **a** Data acquired showing particles and **b** representative image depicting best 2D particles picked for map generation. **c** Cryo-EM density resulting from the helical reconstruction of the NdasCDO filament. On the right side, filament is rotated and subunits shown as cartoons. Box highlights the functional dimer or dimers between A and B subunits. **d** View of the functional dimer of dimers between A and B subunits. The FMN cofactor (teal, sticks) is flanked by residues from both A subunits and one of the B subunits. **e** Surface representation of the refined models for subunits A (left) and B (right).

with cFP and since this substrate has no ionizable side chains, it is not contributing to the effects observed. Structural data led us to hypothesize one of these ionizable groups observed as contributing to $pK_{a1}$ could be Y36 (subunit B). $pK_{a2}$ could be reporting on the FMN cofactor as no other ionizable groups with predicted $pK_a$ values > 8.0 are present in the vicinity of FMN. A proposed $pK_a$ of ~8.3 for the flavin semiquinone in vanillyl-alcohol oxidase has been reported[33], and future studies on the precise nature of this ionizable group as well as flavin intermediates formed during turnover can be performed to better ascertain group(s) giving rise to the pH-rate profiles observed here.

After pH and temperature studies, the best reaction conditions to monitor the NdasCDO-catalysed reaction were established (30 °C at 50 mM Tris, pH 9.0 with 200 mM NaCl), and then employed for subsequent substrate scope, viscosity studies, solvent kinetic isotope effects and proton inventory studies (discussed below).

## Oligomeric state

Purified NdasCDO displayed a large molecular weight, as previously observed for other CDO for which a preliminary characterization was performed[18,20]. We performed mass photometry experiments (Fig. 2d) to determine a salt-dependent oligomerization process, in which higher salt concentrations led to a more homogeneously distributed population of NdasCDO with a MW of 205 ± 47 kDa. Functional enzyme contains subunit A (monomer MW = 22 kDa) and subunit B (monomer MW = 11 kDa) at an unknown ratio. These results suggest NdasCDO is a filament, as observed in a homologue enzyme[21].

## NdasCDO is a filament enzyme

To gain insights into the structural organisation of NdasCDO, we performed helical reconstruction from the cryo electron microscopy data

collected from a cryo vitrified grid containing ordered NdasCDO filaments (Fig. 3a, b) to resolve a 3.0 Å resolution map. This yielded a high-quality map, to which the atomic model containing both A and B subunits was fitted. This model was generated using the AlphaFold2 model for the individual monomers of subunit A and B (Supplementary Fig. 5).

The filament was composed of alternating repeating units of homodimers of NdasCDO subunit A (NdasCDO-A) and homodimers of NdasCDO subunit B (NdasCDO-B), such that the interface between the two homodimers constitutes the heterotetrametric form of the molecule representing the biologically relevant asymmetric unit (Fig. 3e). Continuous filament is formed by shifting the homotetramer along the screw axis with helical rise of 45.45 Å, helical twist of 132.37° and overall helical pitch of 123.6 Å (Fig. 3c).

Subunit A: The structural fold of NdasCDO-A resembled that of the flavin-dependent nitro reductase (NTR) family of proteins[34]. NdasCDO-A was composed of four anti-parallel beta strands surrounded by five alpha helical elements (Fig. 3d). The largest alpha helix α4 (Supplementary Fig. 6a) lies parallel to the β-sheet and carries the highly conserved cysteine Cys121. Within the NdasCDO-A map, evident compatible density is found for the FMN cofactor stemming from the Cys121 residue (Supplementary Fig. 6d). The modelled FMN cofactor clearly fits the density whose 8α carbon lies at distance of ~1.7 Å to the thiol group of Cys121, confirming the covalent linkage with FMN. The α4 helix along with covalently linked FMN is the central interaction site for the homodimer formation (Supplementary Fig. 6b). Surrounding the α4 helix, two other alpha helices - α1 and α2 - from each monomer stack against the α4 and together form the central helical core interaction of the dimer. The C-terminal region from β4 (region 172–195) forms an extended region, carrying a small $3_{10}$ helix and a β-strand that sits parallel to the β1 of the second monomer. Altogether the intact

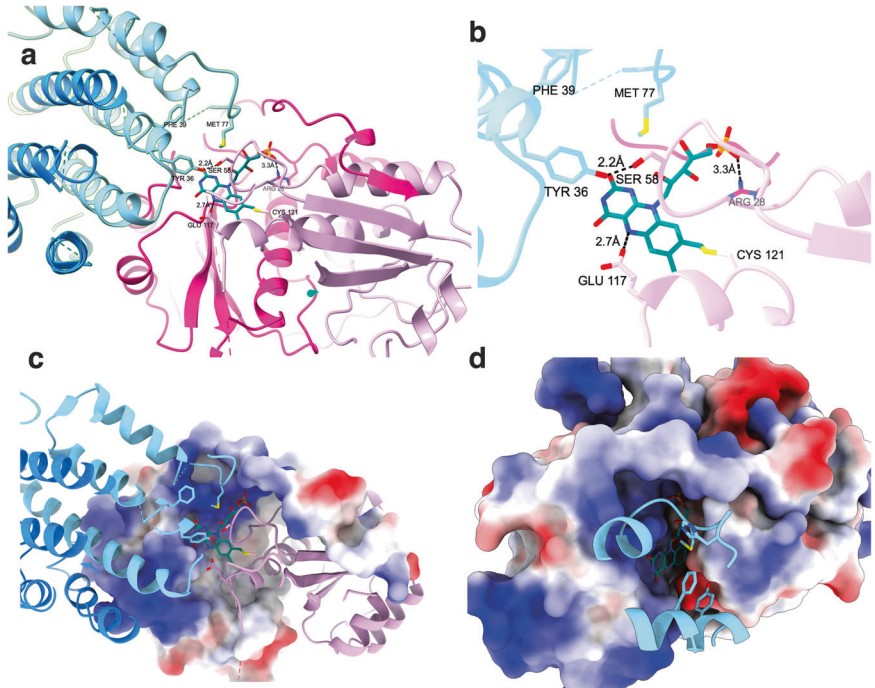

**Fig. 4 | FMN binding site and active site environment. a** Overview of NdasCDO-A (pink) and NdasCDO-B (blue) subunits, FMN (teal) is covalently linked to Cys121, and residues in the vicinity of the cofactor are shown. Supplementary Fig. 7d depicts continuous density between Cys121 and the FMN cofactor. **b** Key residues interacting with the FMN cofactor are shown. One monomer from B subunit covers the active site, and Met77, Phe39, Tyr36 are shown as sticks. From NdasCDO-A, Glu117 is in close proximity to the N5 of FMN, Ser58 is in close distance to NdasCDO-B Tyr36. From a second NdasCDO-A monomer (dark pink), Arg28 is in a polar

interaction with the phosphate group of the cofactor. **c** One monomer of NdasCDO-A is shown as surface, coloured by electrostatics. The putative substrate binding pocket is therefore covered by Met77 and Tyr36 from NdasCDO-B. **d** Both monomers of NdasCDO-A are shown as surface, coloured by electrostatics, and key regions from NdasCDO-B are shown as cartoons. FMN is in a deep pocket, mostly surrounded by positively charged and hydrophobic residues. For all panels, FMN is in teal, one subunit A is in dark pink and the other on light pink, one B subunit is shown in light blue, and the other in darker blue.

homodimer forms the active site region for the FMN cofactor and potentially binds the CDP substrate. Comparison to the only known other CDO structure[21], AlbAB, highlighted a high degree of similarity between each A subunit (Supplementary Fig. 7a) with an RMSD of 0.809 Å across 158 pruned atom pairs (as calculated by MatchMaker in UCSF ChimeraX, version 1.7.1[35]). This is not unexpected given that these enzymes are from the same family and further supports the work by Giessen et al.[21].

Subunit B: NdasCDO-B is an α-helical protein composed of 4 antiparallel helices and, together with the second monomer, forms a dimeric 8-helical bundle structure (Fig. 3d, e, Supplementary Figs. 5 and 6a). Residues from both monomers carry a heptad repeat element pattern which, through extensive inter-molecular interactions, forms the interlock between the two monomer helices to create a knob-into hole structure. Helices α1, α2 and α3 from one monomer together with α4 from second monomer create a heterotetramer interface with a subunit from the NdasCDO-A homodimer. Again, close resemblance was observed between NdasCDO-B and AlbB[21] (Supplementary Fig. 7b), but to no other known protein.

The FMN cofactor is covalently attached to Cys121 from subunit A through an 8α-S-cysteinyl linkage and sandwiched between subunits A and B (Fig. 4a). Covalent attachment to subunit A and to this cysteine residue specifically was confirmed by protein mass spectrometry (Supplementary Fig. 2). Giessen and colleagues highlighted key conserved residues from both subunits that could be pivotal in FMN binding[21]. In comparison, the same residues were observed in NdasCDO including R32, N59 and S145 from NdasCDO-A and E32, P33, Y36, and R40 from NdasCDO-B. Whilst R32 is in close proximity to one of the C2 carbonyl groups from FMN's isoalloxazine ring, R28, R29 and R181 also participate in polar interactions with the phosphate group of

FMN, (Fig. 4b and Supplementary Fig. 6c). However, variants harbouring a truncated N-terminus (NdasCDO$_{\Delta 2-17}$ and NdasCDO$_{\Delta 2-39}$), specifically NdasCDO$_{\Delta 2-39}$ which removed R28, R29 and R32 retained catalytic activity. NdasCDO$_{\Delta 2-17}$ was comparable to wild type in the formation of singly and doubly oxidized products, and NdasCDO$_{\Delta 2-39}$ could catalyse CDP oxidation, albeit with much lower yields. It is likely that this truncation impairs catalytic turnover, given polar contacts to optimally position the FMN cofactor. Nevertheless, these truncations demonstrated that the flexible N-terminus of CDO-A was not required for function.

A DALI search revealed the closest homologue to NdasCDO-A was the nitroreductase, NfsA[34] (Supplementary Fig. 8). Catalytic Ser41 of NfsA is crucial for function however, mutating the equivalent serine residue to alanine (NdasCDO-A$_{S58A}$) did not lead to complete loss of function (Supplementary Fig. 9). Nevertheless, NdasCDO-A$_{S58A}$ had impaired $k_{cat}$ and $k_{cat}/K_M$ and, given the proximity to Tyr36-B (NdasCDO-B), Ser58 could be important in the catalytic mechanism of the reaction. In addition to catalytic relevance, Tyr36-B could also participate in stabilising interactions with the flavin isoalloxazine ring through its aromatic side chain. This would be reminiscent of another flavoenzyme, monoamine oxidase, which also displays the same chemical linkage to flavin cofactor[36]. Furthermore, Glu117, from one of the monomers of NdasCDO-A, is within hydrogen bonding distance from the N5 of FMN. However, Glu117 is 7 Å from Ser58-A or Tyr36-B, too far for a direct interaction but nevertheless in a position that could lead to interactions with CDP substrates and their side chains to allow proton transfer with the cofactor. Together Tyr36-B, Phe39-B and Met77-B could limit access to the substrate binding site and FMN, as the bulky side chains obstruct the active site (Fig. 4c). The FMN and substrate binding pocket are in a deep cavity, mostly surrounded by positively

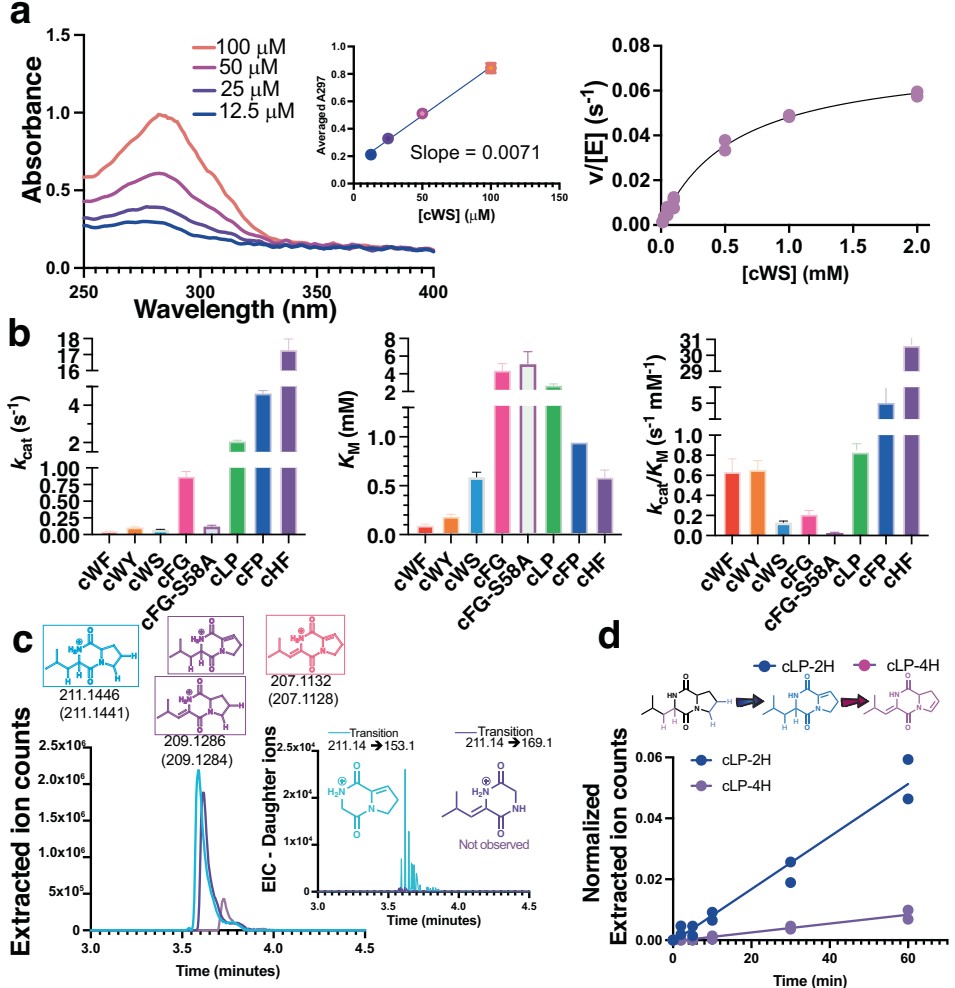

**Fig. 5 | Substrate scope of NdasCDO. a** Overview of workflow to test different CDP substrates and determine kinetic parameters. First (left), progress curves were acquired overnight, and after reaction had reached completion final absorbance readings were utilized to generate a calibration curve correlating change in absorbance to amount of product formed (middle). Then, assays under initial rate conditions were carried out to determine substrate scope and preference (right). Only absorbance readings including under 10% substrate conversion into products were considered. **b** Fitted values for $k_{cat}$ (left), $K_M$ (middle), and $k_{cat}/K_M$ (right) are shown with standard error of the fit. All experiments were carried out in three independent experiments and raw data for individual progress curve are available on Supplementary Fig. 3, all Michaelis–Menten plots available on Supplementary Fig. 10. **c** Unique transitions observed by mass fragmentation of cLP-2H oxidation (Supplementary Fig. 13). Inset shows in blue a unique transition when cLP-2H undergoes dehydrogenation first on the proline sidechain, while the species in purple (not detected) depicts a unique transition if cLP-2H underwent dehydrogenation first on the leucine sidechain. Fragmentation was determined as previously described[27]. **d** Reaction with cLP monitored by mass detection, using single reaction monitoring (SIR) channels for substrate, product with 1 oxidation (cLP-2H, blue) or 2 oxidations (cLP-4H, pink). Two biological replicates are shown, data fitted to a linear regression curve. Under these conditions there is a lag for formation of cLP-4H, indicating sufficient levels of cLP-2H needed to accumulate before a second dehydrogenation reaction could take place. Supplementary Fig. 12 depicts a progress curve for cFP oxidation, for which the reaction for the first oxidation is faster and the lag before cFP-4H formation is more pronounced.

charged and hydrophobic residues, and further shielded from solvent by Tyr36-B, Phe39-B and Met77-B (Fig. 4d).

## Substrate scope of NdasCDO

NdasCDO is a promiscuous enzyme, accepting multiple CDP substrates[20,27]. Assay conditions and procedure to determine extinction coefficients employed are available in the supporting information (example shown in Fig. 5a and more in Supplementary Fig. 3). $K_M$ values were between 0.08 mM (cWF) and 4.2 mM (cFG), indicating overall poor affinity for CDP substrates (Fig. 5b and Supplementary Fig. 10). Surprisingly, cHF was the best substrate ($k_{cat}/K_{M\text{-}cHF} = 30.6 \pm 0.1$), mostly due to a $k_{cat}$ effect. The CDPS enzyme from part of the same biosynthetic gene cluster as NdasCDO is a natural producer of cWF, a substrate with overall poor efficiency, and in the

same magnitude of cLP, demonstrating a non-strict preference for aromatic amino acid side chains.

Other CDPs were tested as substrates (cWG, cWW, cFF, cHP), and were successfully oxidized (verified by LC-MS in Supplementary Fig. 11), however a combination of high $K_M$ values, slow reactions, substrate solubility and small changes in absorbance due to product formation did not allow determination of kinetic parameters for all substrates tested. LC-MS assays revealed cWG underwent slow oxidation with little product formed after 2 h reaction.

We tested cWS as a substrate as the acceptance of substrates with polar side chains was undetermined. Furthermore, cWS is an interesting peptide with antibiofilm activity against *Pseudomonas aeruginosa*[37]. With purified reaction components, NdasCDO can catalyse single and double dehydrogenation, although the product cWS-2H

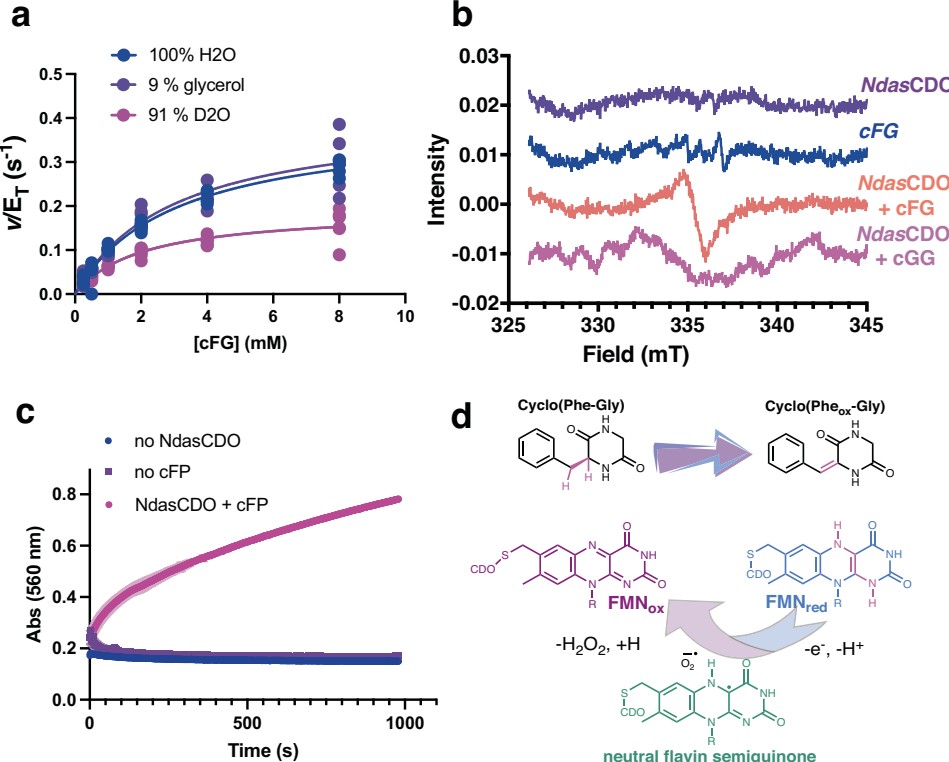

**Fig. 6 | Rate limiting protonation steps and radical formation on the NdasCDO-catalysed reaction. a** Solvent kinetic isotope effects with 91% $D_2O$ and with 9% glycerol as a viscosity control yield $^{D2O}V/K_{M\text{-}cFG} = 1.4 \pm 0.2$, and $^{D2O}V = 2.1 \pm 0.1$. Data for six replicates are shown. Line is a fit to Eq. (4). **b** Continuous wave EPR spectra acquired in samples frozen in liquid nitrogen with 5 mM cFG (orange) or cGG (pink) and 100 μM NdasCDO. Controls with cFG only (blue) or NdasCDO only (purple) are also shown. **c** Coupled reaction with hydrogen peroxidase and ABTS (2,2'-Azinobis [3-ethylbenzothiazoline-6-sulfonic acid]-diammonium salt), showing hydrogen peroxide formation. **d** Summarized catalytic cycle depicting a potential neutral flavin semiquinone intermediate.

accumulated after an overnight reaction, indicative that the second dehydrogenation was not favoured. Prior work with cell cultures and NdasCDO failed to observe cWS as a feasible substrate[20], and we hypothesize this could be due to 1- the high $K_M$ value for cWS (0.8 mM), and 2- the higher complexity and suppression effects observed when carrying out LC-MS assays with complex mixtures for CDP detection[38].

### Studies on second oxidation

Most CDP substrates tested can be doubly oxidized by NdasCDO. We, therefore, performed a time course experiment monitoring, by mass detection, the appearance of the first oxidation product followed by the second oxidation product (Fig. 5c). Progress curves were used to monitor the disappearance of cLP (substrate) and the appearance of cLP-2H (first product) and cLP-4H (second product) (Fig. 5d). cLP was completely converted into cLP-4H following an overnight reaction where an initial lag phase was noted to accumulate sufficient concentrations of cLP-2H for catalysis. A similar pattern was observed for cFP (Supplementary Fig. 12). We propose NdasCDO is releasing a CDP-2H product, which then must rebind in a different orientation so that second oxidation can occur. Overall, substrates containing one tryptophan were poor candidates for formation of doubly oxidized products, while substrates containing Phe and other small hydrophobic side chains were more efficient substrates. Modelling CDP substrates into a substrate binding pocket in the vicinity of the FMN cofactor suggests that substrates such as cFG sample fewer conformations than substrates that can undergo two oxidation events (cFP, cWF, cWS, cWY, cLP, cHF), and more conformations are sampled by substrates that contain tryptophan, in agreement with their lower $k_{cat}/K_M$ values (Supplementary Fig. 14).

### Rate limiting steps and mechanism for the NdasCDO-catalysed reaction

To probe the rate limiting nature of proton transfer steps, we determined SKIEs with cFG as a substrate. Fit to Eq. (4) yielded $^{D2O}V/K_{M\text{-}cFG} = 1.4 \pm 0.2$, and $^{D2O}V = 2.1 \pm 0.1$. Proton inventories were complex and the best fitted model accounts for one transition state proton and one reactant state proton contributing to the observed $^{D2O}V = 2.1 \pm 0.1$ (Supplementary Fig. 15). Due to complex nature of the enzyme filament, the fact that the fitted fractionation factors assume the $^{D2O}V$ is an intrinsic isotope effect (when there is no evidence supporting or discarding this assumption), we do not discuss this further in terms of mechanism, and future studies focused on the catalytic and chemical mechanism of this reaction are needed to establish the nature of this complex effect.

To rule out differences in viscosity between $H_2O$ and $D_2O$ giving rise to the observed SKIEs, we performed control experiments with 9% glycerol, as this yields a relative viscosity akin to $D_2O$. No viscosity effect at this concentration was observed (Fig. 6a).

Additionally, glycerol, sucrose and PEG were used as viscogens to determine whether diffusional steps are contributing to $k_{cat}/K_{M\text{-}cFP}$ and $k_{cat}$ (Fig. 6a). Glycerol and sucrose are acting as microviscogens, while PEG 8 K acts as a macroviscogen[39]. No effect was observed with increasing concentrations of neither viscogen (Supplementary Fig. 16, slopes equal to $0.1 \pm 0.1$). We conclude diffusional steps included in substrate binding and product release as well as large conformational changes the filament enzyme might be undergoing during catalysis do not significantly limit catalytic turnover[39].

As many flavin dependent enzymes have radical reaction intermediates, we used EPR spectroscopy to test whether this was the case

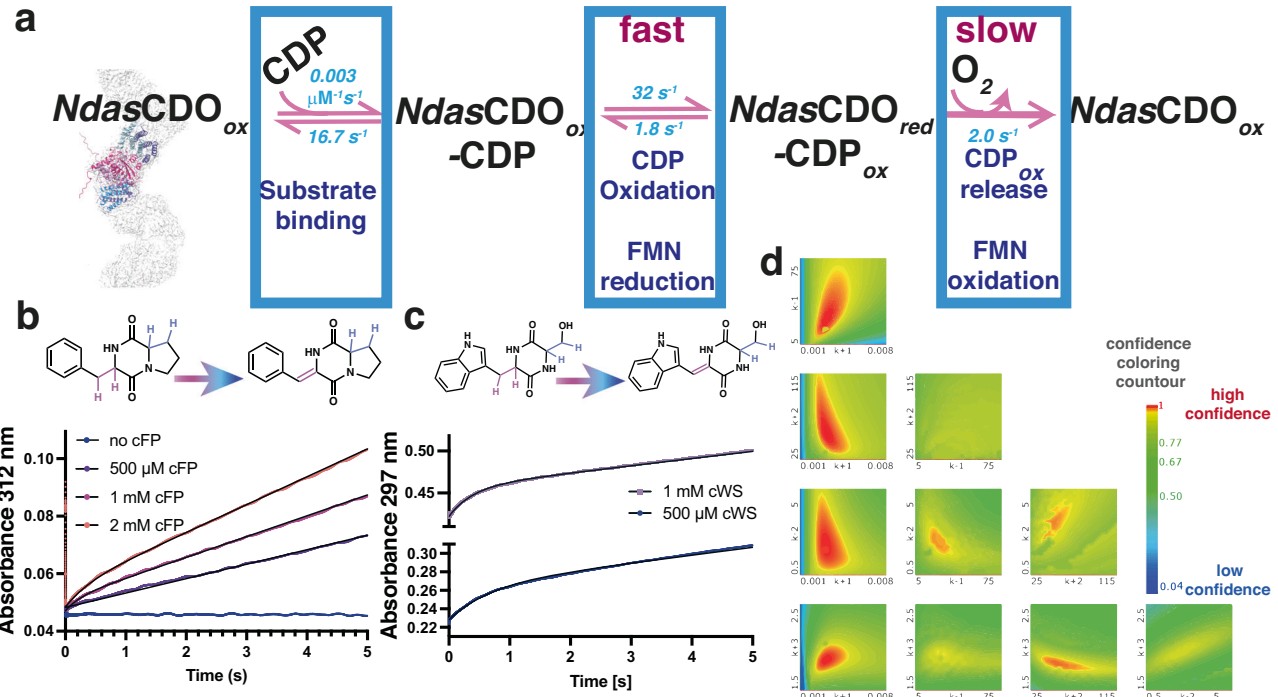

**Fig. 7 | Catalytic cycle for the NdasCDO-catalysed reaction. a** Model used to fit multiple turnover experiment with substrate cFP, displaying best fitted values. Boxes depict macroscopic steps, as the signal observed originated from the formation of cFP-2H. The last step in the model includes product release and FMN re-oxidation so next catalytic cycle can occur. **b** Raw data for multiple turnover experiment with 8.5 µM NdasCDO and varying concentrations of cFP (0.5, 1 and 2 mM), monitored at 312 nm. **c** Raw data for multiple turnover experiment with 8.5 µM NdasCDO and varying concentrations of cWS (0.5, and 1 mM), monitored at 297 nm. For **b** and **c**, in the time frame of the experiment we expect a single oxidation to take place, data are depicted as the average of three individual replicates and the black line is a fit to Eq. (7). **d** Results from data fitting of results shown on panel **b** using Kintek Global Explorer and output from the three-dimensional error analysis performed using Fitspace (included in the Source Data file). High confidence values for the fits are depicted in red, bright yellow boundaries represent the limits of the confidence interval for each rate constant when varied in function of every other rate constant in the mechanism proposed. Table 1 summarizes the values obtained after fitting.

for NdasCDO. We observed a radical formed solely in the presence of enzyme and the cFG substrate, while no radical formation was seen in the enzyme alone or with cFG under identical experimental conditions (Fig. 6b). We also did not observe radical formation with NdasCDO and cyclo-Glycine-Glycine (cGG), and therefore conclude the radical observed is exclusively formed as reaction takes place. Figure 6d depicts a neutral flavin semiquinone intermediate, which is in agreement with our data, however, the precise nature of this radical remains to be determined by additional work.

In the CDP from *Streptomyces albulus*, hydrogen peroxide is formed during turnover[18]. To confirm $H_2O_2$ formation in the NdasCDO-catalysed reaction we used a coupled assay with hydrogen peroxidase and ABTS (2,2′-Azinobis [3-ethylbenzothiazoline-6-sulfonic acid]-diammonium salt). An increase in absorbance at 560 nm only in the presence of NdasCDO and cFP, consistent with $H_2O_2$ formation, was observed (Fig. 6c).

To determine whether the first turnover of the reaction is slower than subsequent turnovers (burst of product formation), indicative of a step after the chemical step under evaluation being rate limiting, we carried out multiple turnover experiments under pre-steady state conditions (Fig. 7a). A clear burst was observed for two distinct substrates (Fig. 7d), with similar observed rate constants for the exponential phase when fitted analytically at 1 mM for each substrate $k_{obs\text{-}cFP} = 2.6 \pm 0.1\,\text{s}^{-1}$ (Fig. 7c), $k_{obs\text{-}cWS} = 4.2 \pm 0.1\,\text{s}^{-1}$ (Fig. 7b). Care must be taken comparing both substrates, as they likely have different affinities for the enzyme, but nevertheless rates for the exponential burst phase with both substrates are in the same order of magnitude. Table 1 summarizes the values obtained after fitting data on Fig. 7.

Data acquired with cWS possesses higher signal amplitude, and therefore burst is more pronounced, but absorbance at 297 nm is complex given contributions from the FMN cofactor, the enzyme and the substrate. Alternatively, cFP dehydrogenation leads to increase in absorbance at 312 nm, which is distinctive given cFP has no absorbance at this wavelength, the contribution from the enzyme is less significant, enabling data analysis using a more complex model and numerical fitting. Our three step model accounts for substrate binding, CDP oxidation and FMN oxidation/product release. Because we don't have observable signals to discriminate individual rate constants for FMN oxidation and product release in individual steps, they were combined as a single irreversible apparent rate constant. However, the absence of viscosity effects and observation of a sizeable solvent kinetic isotope effect on $k_{cat}$ point towards a step coupled to FMN re-oxidation as rate limiting, instead of product diffusion from enzyme. This can be directly linked to FMN re-oxidation or be due to a proton-transfer-linked

**Table 1 | Rate constants obtained after fitting multiple turnover experiment**

|          | Best fitted value     | Lower boundary[a] | Upper boundary |
|----------|-----------------------|-------------------|----------------|
| $k_1$    | 0.0031 ± 0.0001       | 0.0025            | 0.0038         |
| $k_{-1}$ | 16.8 ± 0.8            | 18.2              | 38.6           |
| $k_2$    | 32.0 ± 0.5            | 49.2              | 75.3           |
| $k_{-2}$ | 1.8 ± 0.1             | 1.6               | 2.2            |
| $k_3$    | 2.0 ± 0.1             | 1.8               | 1.8            |

[a]Boundaries were calculated with $\chi^2 = 0.95$, and $k_{-3}$ was fixed at zero, since product release was considered irreversible under multiple turnover conditions.

conformational change, as proposed for the phosphoribosyl-ATP Pyrophosphohydrolase/Phosphoribosyl-AMP cyclohydrolase (AbHisIE) from *Acinetobacter baumanii*[40].

### Importance of filamentation for structure and catalysis

Oxidized cyclic peptide natural products such as phenylahistin[6] and albonoursin[19] have anticancer activity, and the importance of dehydroamino acid residues in bioactive natural products has been established[41]. Therefore, CDO enzymes can be useful biocatalysts, provided they are better understood in terms of substrate selection and properties.

Performing dehydrogenation en masse by utilising a CDO biocatalyst would maximise atom economy whilst minimising unwanted side products and waste from this selective reaction. Previous attempts at mutating enzymes that catalyse reactions via radical intermediates successfully expanded the overall substrate scope[42] for improved biocatalysts. Furthermore, previous work by our group[38] and others[43] established that the biosynthetic partner of CDOs, cyclodipeptide synthases, were amenable to active site manipulation for expanded substrate acceptance[38], and that these enzymes were capable of utilizing unnatural amino acids[44,45]. Other tailoring enzymes have been described, which catalyse prenylation[46] as well as other complex reactions[47]. Thus, our work in better understanding the structure and kinetics of NdasCDO is a necessary step towards combining these strategies to generate novel cyclodipeptide analogues.

Here we report details on the mechanism and structure of a cryptic member of the cyclodipeptide oxidase family. In agreement with recent findings by Giessen et al.[21], NdasCDO was observed as a high molecular weight oligomer whose length is dependent on overall salt concentration. NdasCDO is an enzyme from the halophile *Nocardiopsis dassonvillei*, and therefore the environment naturally occupied by *N. dassonvillei* could be conducive to filament formation and generation of oxidized CDPs. Prior work with ethyl acetate extracts of this actinobacteria demonstrated biofilm-inhibitory activity, although the precise identity of the natural product(s) giving rise to this effect remains to be determined[48]. Previous research discussed the inactivation of CDOs when either subunit was absent, and we demonstrate structural features underlying this reliance on a functional complex. Filament formation is fundamental for enzymatic activity, as our structural work shows residues from subunits A and B are important to interact with substrates and the covalently bound FMN cofactor.

Whilst NdasCDO-A harbours high structural similarity to nitroreductases, our work highlights that the active site of CDOs differs from these enzymes, as it does not rely on the same catalytic mechanism and structural features. Instead, NdasCDO is a distinct flavoenzyme[30], operating optimally in high salt and high pH conditions without the need for additional cofactors, in a reaction mechanism limited not by the complex three-dimensional arrangement of the protein, but by cofactor regeneration. Combining our kinetic and structural data we put forward a chemical mechanism involving a neutral flavin semiquinone radical intermediate, in an initial fast phase in which Ser58-A and Tyr36-B play a crucial role. Despite acting in many substrates, NdasCDO has a broader scope when catalysing a single oxidation reaction, rejecting large amino acid side chains such as tryptophan, and being more restrictive for substrates in a second oxidation reaction.

The active site of NdasCDO comprises residues from two A subunits and one B subunit, providing functional rationale for its occurrence as a filament. Furthermore, filamentation could aid NdasCDO in evading degradation during starvation or stress[22] – both environmental conditions that *Nocardiopsis dassonvillei* thrives in ref. 48. Filament enzymes are emerging as an important concept in the regulation of protein function[49], and therefore our work sheds light into how structure and function are amalgamated to control activity in one of these complex systems.

## Methods

### Chemical reagents and general methods

Buffers, salts, and other common chemicals were purchased from Merck or Fisher Scientific and used without further purification. All cyclodipeptide substrates (except for cWS, which was synthesized chemically as described in the Supplementary Note 1) were purchased from Bachem and used without further purification. All spectrometric experiments were performed in independent experiments in triplicate using a BMG Labtech POLARstar Omega plate reader. All experiments in replicate were from individual measurements, and not from repeated measurements on the same sample. Errors were propagated following standard formulas for uncertainty propagation, including for all $k_{cat}/K_M$ values[50].

### Expression, purification, and mutagenesis of NdasCDO

Genes encoding Ndas1146 (Uniprot: D7B1W6) and Ndas1147 (Uniprot: D7B1W7) from *Nocardiopsis dassonvillei* were cloned into a pRSFDuet-1 vector by Genscript (protein sequences available in the Supplementary information "supporting methods", and plasmid map on Supplementary Fig. 17). Before further use, the plasmid was transformed into commercially available *Escherichia coli* (*E. coli*) DH5α cells (NEB). The DNA was extracted following the QIAprep Spin Miniprep Kit (Qiagen) and sequenced by DNA Sequencing & Services (MRC I PPU, School of Life Sciences, University of Dundee, Scotland, www.dnaseq.co.uk) using Applied Biosystems Big-Dye Ver 3.1 chemistry on an Applied Biosystems model 3730 automated capillary DNA sequencer. Mutants of NdasCDO were created by site-directed mutagenesis based on the NEB Q5 site-directed mutagenesis kit (Supplementary Table 1 for primer details).

For protein expression, purified plasmid was transformed into *E. coli* BL21(DE3) competent cells (NEB). Cells were grown at 37 °C at 180 rpm until an optical density ($OD_{600}$) between 0.6 and 0.8 was reached. Protein expression was induced upon the addition of IPTG to a final concentration of 500 μM. Cells were incubated overnight at 25 °C before being harvested via centrifugation (6774×*g*, 15 min, 4 °C). Cells were resuspended in lysis buffer consisting of 100 mM Tris pH 8, 200 mM NaCl, 20 mM imidazole, 10% (v/v) glycerol and stirred to homogeneity in the presence of lysozyme (10 mg) and DNase I (1 mg) at 4 °C. Cells were lysed at high pressure using a cell disrupter at 30 kpsi and the lysate was clarified via centrifugation (48,000 × *g*, 30 min, 4 °C). The resultant supernatant was filtered through a 0.8 μm membrane before being loaded onto 5 mL HisTrap FF nickel column (GE Healthcare) pre-equilibrated in lysis buffer. Proteins were eluted in steps of 10%, 20% and 100% elution buffer containing 100 mM Tris pH 8, 200 mM NaCl, 300 mM imidazole, and 10% (v/v) glycerol. Fractions of interest were pooled together for dialysis into 50 mM Tris pH 8, 200 mM NaCl, 10% (v/v) glycerol overnight at 4 °C. Following dialysis, the protein was concentrated using an Amicon Stirred Cell (Merck) equipped with a 100 kDa molecule weight cut off (MWCO) ultrafiltration disc (Millipore). To concentrate to a final smaller volume, the protein was transferred to a Vivaspin® 6 centrifuge unit with a 100 kDa MWCO PES membrane (Sartorius).

### Melting temperature investigations of NdasCDO

Differential scanning fluorimetry was performed in independent experiments conducted in triplicate using a Quantstudio real time thermal cycler (Thermofisher). SYPRO™ Orange Protein Gel Stain (Thermofisher) was used to a final concentration of 10x with 50 μM NdasCDO in 20 μL. Buffer and salt conditions were varied using a series of conditions prepared in an in-house stability screen, and the temperature was ramped from 25 to 95 °C over 90 min. Melting temperatures were calculated by fitting the data to the Boltzmann sigmoidal function (Graphpad Prism).

## Mass photometry

Mass photometry was performed using a OneMP mass photometer (Refeyn Ltd). Microscope slides (70 ×26 mm) were cleaned first with 100% (v/v) isopropanol followed by pure MilliQ water. The procedure was repeated twice and then slides were air dried using a pressurized air stream. Silicon gaskets to hold the sample drops were cleaned in the same manner as described for the slides. All preparation were done prior to immediate start of the experiment. All data were recorded using AcquireMP software. The instrument was calibrated using NativeMark protein standards in the same sample buffer (10 mM Tris pH 9.0, 200 mM NaCl, 5 mM DTT) prior to sample measurements. Stock protein sample was diluted to final concentration of 100 nM. To find focus, 8 µL of buffer was pipetted into a well and the focal position was identified and locked using the autofocus function. Then to acquire sample measurements, 2 µL of 100 nM concentration of protein was pipetted into 8 µL of the buffer and mixed carefully. Mass photometry movies were then recorded for 120 s. The data were analysed using the DiscoverMP software and generated mass distribution histograms.

## Cryo-electron microscopy (Cryo-EM) for determination of NdasCDO structure

**Sample preparation for cryoEM.** Prior to preparing grids, purified NdasCDO complex was dialysed against 20 mM Tris pH 9, 200 mM NaCl, 5 mM DTT for 2–3 h. The dialysed protein was diluted to 4 µM final concentration and spun at 6010 g on an Eppendorf table-top centrifuge for 5 min to remove any aggregates. Grids of the NDS protein were then prepared by applying 4 µL of 4 µM protein to a glow-discharged (1 min at 0.1 mbar, 35 mA Quorum technologies SC7620) Quantifoil R2/1400 mesh Cu/Rh holey carbon grids. The grids were then vitrified by plunging into liquid ethane using FEI Vitrobot Mark IV. Before plunging, grids were blotted for 3 s with the blot force of 3.5, and the climate chamber was maintained at 4 °C and 100% humidity. The grids were subsequently clipped and stored in liquid nitrogen until data collection.

**Data collection, processing, and model building.** Initial screening of NdasCDO grids were carried out in our in-house Glacios microscope equipped with Falcon4i camera (University of Dundee, CryoEM facility). Best grids with well-defined ice and good distribution of filaments were then transported to eBIC cryo electron microscopy facility located at Harwell Diamond. Data were then collected using a 300Kv Titan Krios cryo electron microscope equipped with Gatan K3 direct electron detector. EPU software was used to select targets and acquire movies in super resolution electron counting mode at a magnification of 105,000x and calibrated sampling pixel size of 0.831 Å/pixel after 2X binning. In total, 12,900 movies were collected with 50 frames per movie stack with a total dose of 33.44 e/Å$^2$ and exposure time of 1.79 s per movie stack using aberration free image shift (AFIS). More information can be found in Supplementary Fig. 18 and Supplementary Fig. 19.

Structural reconstruction using helical processing methodology was carried out using cryoSPARC suite (v4.4.1)[51]. Movies were imported into Cryosparc and using patch motion correction, frames were aligned, and both beam induced motions correction and electron dose weighting were performed. CTF estimation using CTFFIND4 was carried out and those micrographs with CTF fit worse than 4.8 Å were discarded, which resulted in 9031 images. Particles were initially picked and optimised on a subset of 200 images using Filament tracer with filament diameter of 120 Å, which was subsequently extended to all the images. A total of 1,800,000 particles were picked and extracted with a box size of 340 pixels. The extracted particles were then inspected, filtered for any ice contamination and artifacts, and subjected to three rounds of 2D classification. The best 2D classes were selected, resulting in 849600 particles and a proportion of particles (180,000) from the classes were used for initial Ab-initio model

generation. Subsequently, all 849,000 particles and the ab-initio model were used as input for a Helix Refine job. The map generated from the Helix Refine job had a global resolution of 3.6 Å. Helical parameters including helical rise and twist were searched within the reconstructed volume and the initial estimates were input for the Helix refine job to yield a final estimated helical twist of 132.3° and a helical rise of 45.45 Å. The particle stack was then subjected to Global and local per particle CTF refinement before performing second round of Helical refinement which improved the map to 3.4 Å resolution. A local refinement was carried out using a mask covering two dimers of NdasCDO subunit A and two dimers of NdasCDO subunit B, yielding a final map of 3.07 Å global resolution. The final map was filtered and sharpened using B-factor of 180 Å$^2$.

AlphaFold2[52] was used to generate an initial model of the individual NdasCDO subunit A and B monomers. The models were then rigid body docked into the filament map and respective oligomers were generated. The structure was refined with rounds of model building in Coot, fitting with adaptive distance restraints in ISOLDE (v1.6)[53] and refinement with Phenix (v1.20.1-4487)[54] real-space refinement. Figures were generated with the PyMOL Molecular Graphics System, version 1.8[55] and UCSF ChimeraX[35].

## Electron Paramagnetic Resonance (EPR) spectroscopy

5 mM cFG or 5 mM cGG were added to 100 µM NdasCDO and the reaction mixtures were frozen using liquid nitrogen. Controls of 5 mM cFG and 100 µM NdasCDO as well as a buffer-only sample (for background-subtraction) were also prepared. Continuous wave EPR spectra were obtained at 120 K with a Bruker EMX 10/12 spectrometer running Xenon software and equipped with an ELEXSYS Super Hi-Q resonator at an operating frequency of ~9.50 GHz with 100 kHz modulation. Temperature was controlled with an ER4141 VTM Nitrogen VT unit (Bruker) operated with liquid nitrogen. CW spectra (757 scans per sample) were recorded using a 20 mT field sweep centred at 336 mT, a time constant of 5.12 ms, a conversion time of 20.02 ms, and 1000 points resolution. An attenuation of 23.0 dB (1 mW power) and a modulation amplitude of 0.2 mT were used. CW spectra were phase- and background-corrected using the Xenon software.

## pH-rate profiles

Prior to the execution of pH-rate profiles, enzymatic stability tests were performed. NdasCDO was diluted to 10 µM in H$_2$O or 200 mM of a mixed buffer containing MES, Tris, CAPS and KCl at either pH 6 or 11. NdasCDO, at a final concentration of 0.1 µM, was then added to a reaction mixture containing 50 mM Tris pH 8 and 200 mM NaCl. cFP was used at saturating concentrations to verify that NdasCDO retained activity following incubation at both low and high pH.

For pH profiles, the mixed buffer system was prepared to range from pH 6–11 in increments of 0.5 pH units. A final concentration of 200 mM of each buffer was used whilst cFP substrate concentration was varied. NdasCDO was added last to a final concentration of 0.1 µM and all reactions were carried out in independent triplicate measurements.

## Temperature-rate profiles

Firstly, to determine whether NdasCDO was stable at the temperatures used in this study, NdasCDO was incubated at the extremes of the experimental temperature range and subsequently assayed at 298 K. This confirmed no loss of enzymatic activity occurring upon incubation for 30 min.

For temperature investigations, cFP was used at saturating conditions in 50 mM Tris, pH 9 for optimal activity as confirmed by the previous pH-rate profile. NdasCDO was added to a final concentration of 0.1 µM and the activity was measured between a temperature range of 298–318 K in 5 K increments, which was dictated by the constraints of the plate reader used.

## Solvent Kinetic Isotope Effects (SKIEs)

Solvent kinetic isotope effects (SKIEs) were explored by plotting saturation curves in $H_2O$ or 91% $D_2O$. Additionally, a control experiment containing 9% glycerol was performed concurrently. Here, cFG was used as a substrate with 0.1 μM NdasCDO in 50 mM Tris pH 9 and 200 mM NaCl. For proton inventory experiments, $D_2O$ was varied from 0 to 90% in increments of 10%. Buffers prepared in $D_2O$ were corrected for pH alterations caused by the presence of $D_2O$ [pD + = pHa (apparent reading from pH metre) + 0.4]. SKIE investigations in $H_2O$ were analysed using Michaelis–Menten analysis to obtain both $k_{cat}$ and $K_M$. SKIE experiments were carried out in six independent experiments, data were reported as mean and standard error of the mean.

## Viscosity studies

Potential viscosity effects were investigated using either sucrose or glycerol as the viscogen. Relative viscosities ($\eta_{rel}$) for different concentrations of glycerol and sucrose were measured by Bazelyansky et al. and listed in the supporting information on Supplementary Table 2.

For viscosity effects, cFG was used as the substrate at varying concentrations with 0.1 μM of NdasCDO in 50 mM Tris pH 9 and 1 M NaCl. Each independent experiment was incubated at 30 °C in triplicate and the absorbance was measured at 297 nm.

## Determination of extinction coefficients

Spectrophotometric assays were used to monitor the oxidation of cyclodipeptides. Initially, the absorbance spectrum (200–1000 nm) was measured upon the addition of NdasCDO (i.e., 0 min timepoint) and then again after 24 h incubation at 30 °C. From here, the difference in absorbance for each respective oxidised CDP was used to create a progress curve. The last 20 points for each concentration were averaged and plotted against the substrate concentration to give a linear trend where the gradient represented the extinction coefficient for the oxidised CDP in absorbance/concentration (M) units. Supplementary Fig. 3 depicts UV difference spectra upon oxidation for different CDP substrates as well as the progress curves to determine the respective extinction coefficients.

## LC-MS Analysis of oxidised cyclodipeptides

*Completion reactions*: samples containing 1 μM NdasCDO and 30 μM of the respective cyclodipeptide were incubated in 50 mM Tris pH 8 and 200 mM NaCl overnight at room temperature. The reaction was quenched upon the addition of cold methanol to a final concentration of 80%. The samples were incubated at −80 °C for 15 min and centrifuged at 18,000 × g for 10 min. The resultant supernatant was dried under nitrogen gas and reconstituted in LC-MS grade water. Raw data are available in Supplementary Fig. 11a–i. Completion reactions were analysed in single replicates, and successful substrates were further investigated as substrates in progress curves and spectrophotometric assays.

All samples except for cWS were analysed using a Waters ACQUITY UPLC liquid chromatography system coupled to a Xevo G2-XS Qtof mass spectrometer equipped with an electrospray ionization (ESI) source. 1 μL of sample was injected onto the appropriate column and ran at 40 °C. All histidine or tryptophan containing compounds were injected onto a HSS-T3 column (2.1 ×100 mm, 1.8 μm, Waters Acquity) whilst non-polar CDPs such as cFP or cLP were run on a BEH C18 column (2.1 ×100 mm, 1.7 μm, Waters Acquity). CDPs were separated from the mixture using a gradient mobile phase from 1% B to 50% B where the two mobile phases consisted of A – 0.1% formic acid in water and B – 0.1% formic acid in acetonitrile at a flow rate of 400 μL min⁻¹. The capillary voltage was set at 2.5 kV in positive ion mode. The source and desolvation gas temperatures of the mass spectrometer were set at 120 °C and 500 °C respectively. The cone gas flow was set to 50 L/hr whilst the desolvation gas flow was set at 1000 L/hr. An MSE scan was performed between 50 and 700 m/z where function 1 employed MS analysis whilst function 2 applied a collision energy ramp from 15 to 30 V to perform MS/MS fragmentation. In addition, a lockspray signal was measured and a mass correction was applied by collecting every 10 s, averaging 3 scans of 1 s each using 50 pg mL⁻¹ of Leucine Enkephalin dissolved in water:acetonitrile (50:50) and 0.1% formic acid as a standard (556.2771 m/z, Waters).

For cWS, samples were subjected to LCMS/MS using an Eksigent Ekspert nanoLC 425 (Eksigent, AB SCIEX) coupled to a Triple ToF 6600 mass spectrometer (ABSCIEX). Dipeptides were injected on a reverse-phase YMC Triart C18 trap column (12 nm, 3 μM, 0.3 ×0.5 mm) for pre-concentration and desalted with loading buffer at a flow rate of 5 μL/min for 1 min. The peptide trap was then switched into line with a YMC Triart C18 analytical column (12 nm 3 μm 0.3 ×150 mm) column. Peptides were eluted from the column using a linear solvent gradient using the following gradient: linear 3–95% of buffer B within 6 min, isocratic 95% of buffer B for 2 min, decrease to 3% buffer B within 1 min and isocratic 2% buffer B for 4 min. The mass spectrometer was operated in positive ion mode and acquired a TOF MS scan for 120 msec of accumulation time from m/z 80–500, followed by three Product Ion scans for the respective reaction product with the same accumulation time and m/z scan range with a cycle time of 0.5 s.

*Time course reactions:* The reaction stock was prepared in 50 mM Tris pH 9 and 200 mM NaCl with the CDP substrate (cFP or cLP) at 25 μM. NdasCDO was added last to a final concentration of 0.1 μM and the reaction was incubated at 30 °C. At each time point, 40 μL were removed and boiled at 100 °C for 5 min then centrifuged to pellet the denatured protein. The supernatant was then analysed via LC-MS as described above. Reactions were performed in three independent experiments.

## Pre-steady state multiple turnover

Concentrations reported are final after mixing 30 μL each sample in a 1:1 ratio in an Applied Photophysics SX20 stopped flow instrument, with a cell of 5 μL, a pathlength of 0.5 cm and with an instrument deadtime of 1 msec. Each experiment was repeated three times, and exponential curves averaged before fitting. Multiple turnover experiments were carried out as follows:

*cFP:* 8.5 μM NdasCDO and varying concentrations of the cFP (0.5, 1 and 2 mM) in 50 mM Tris pH 9 and 200 mM NaCl. Reaction was monitored at 312 nm for 5 s, 10000 datapoints were collected in logarithmic scale. Voltage applied was 384.4 V.

*cWS:* 8.5 μM NdasCDO and varying concentrations of cWS (0.5 and 1 mM) in 50 mM Tris pH 9 and 200 mM NaCl. Reaction was monitored at 297 nm for 5 s, 10,000 datapoints were collected in logarithmic scale. Voltage applied was 400 V. Data were fitted both analytically using Graphpad Prism (exponential equations in the "Data fitting" section below and numerically using Kintek Global Explorer[56].

## Protein mass spectrometry to investigate flavin cofactor

**Trypsin digestion sample preparation.** Protein was digested into peptides by adding trypsin (1:50 w/w protease:protein) and incubating overnight at 30 °C.

**Data acquisition.** Peptides were subjected to LCMS/MS using an Ultimate 3000 RSLC (Thermo Fisher Scientific) coupled to an Orbitrap Fusion Lumos mass spectrometer (Thermo Fisher Scientific) and equipped with FAIMS. Peptides were injected onto a reverse-phase trap (Pepmap100 C18 5 μm 0.3 × 5 mm) for pre-concentration and desalted with loading buffer, at 15 μL/min for 3 min. The peptide trap was then switched into line with the analytical column (Easy-spray Pepmap RSLC C18 2 μm, 15 cm × 75 μm ID). Peptides were eluted from the column using a linear solvent gradient using the following gradient: linear 4–40% of buffer B over 45 min, linear 40–95% of buffer B for 4 min, isocratic 95% of buffer B for 6 min, sharp decrease to 2%

buffer B within 0.1 min and isocratic 2% buffer B for 10 min. The FAIMS interface alternated between −45V and −65V, and the mass spectrometer was operated in DDA positive ion mode with a cycle time of 1.5 s. The Orbitrap was selected as the MS1 detector at a resolution of 60,000 with a scan range of from m/z 375–1500. Peptides with charge states 2–5 were selected for fragmentation in the ion trap using HCD as collision energy. The raw data files were converted into mgf using MSconvert (ProteoWizard) and searched using Mascot with trypsin as the cleavage enzyme with 1 missed cleavage allowed and Flavin mononucleotide [C(17)H(21)N(4)O(9)P with a delta mass of 456.1046] as a variable modification of cysteines as well as oxidation as a variable modification of methionine residues against an in-house database containing 42,354 protein sequences. The mass accuracy for the MS scan was set to 20 ppm and for the fragment ion mass to 0.6 Da.

**Intact protein mass analysis.** The protein sample (20 μL, 1 μM) was desalted on-line through a MassPrep On-Line Desalting Cartridge 2.1 × 10 mm, using a Waters Acquity H-class HPLC, eluting at 200 μL/min, with an increasing acetonitrile concentration (2% acetonitrile, 98% aqueous 1% formic acid to 98% acetonitrile, 2% aqueous 1% formic acid) and delivered to a Waters Xevo G2XS electrospray ionisation mass spectrometer operated with positive polarity in sensitivity mode. Intermittently a lockspray signal using Leucine Enkephalin was measured and a mass correction was applied. An envelope of multiply charged signals was acquired between m/z 500–2500 and deconvoluted using MaxEnt1 software to give the molecular mass of the protein.

### Data fitting
The entire range of the temperature was fitted to an Eyring equation (Eq. (1))[57–59], where $R$ is the gas constant; $T$ is the temperature (K); $T_0$ is a reference temperature value (30 °C or 303.15 K was used); $k$ is the kinetic rate constant or parameter; $\Delta H^{\ddagger}$ is the enthalpy of activation; $k_B$ the Boltzmann constant; $h$ the Planck's constant; $\Delta S^{\ddagger}$ the entropy of activation.

$$\ln \frac{k}{T} = \ln \frac{k_B}{h} - \frac{\Delta H^{\ddagger}_{T_0}}{RT} + \frac{\Delta S^{\ddagger}_{T_0}}{R} \tag{1}$$

pH data were plotted with pH and log kinetic parameter. Data for $k_{cat}$ pH-rate profiles were fitted to Eq. (2) (accounting for one ionizable group in each limb), while data for $k_{cat}/K_M$ were fitted to Eq. (3) (accounting for two ionizable groups in each limb). For Eqs. (2) and (3), C is the pH independent value of the kinetic parameter; $pK_{a1}$ and $pK_{a2}$ are the dissociation constants for ionizable groups. Equation S1 is an alternative to when ionizable groups are close together, and its usage (or not) is discussed in the Supporting Information.

$$y = \log \left( \frac{C}{1 + \frac{10^{-pH}}{10^{-pKa1}} + \frac{10^{-pka2}}{10^{-pH}}} \right) \tag{2}$$

$$y = \log \left( \frac{C}{1 + \left( \frac{10^{-pH}}{10^{-pKa1}} \right)^2 + \left( \frac{10^{-pka2}}{10^{-pH}} \right)^2} \right) \tag{3}$$

When calculating SKIEs, $k_{cat}$ and $K_M$ were obtained from the Michaelis–Menten curves in $H_2O$. The 91% $D_2O$ Michaelis–Menten curve was fitted to the following equation:

$$Y = (k_{cat}*[S])/\left( K_M*(1 + F_i*E_{V/K}) + (1 + F_i*E_V) \right) \tag{4}$$

Where $F_i$ is the fraction $D_2O$, and $E_{V/K}$ and $E_V$ are the isotope effects -1 on $k_{cat}/K_M$ and $k_{cat}$ respectively.

Fitted values for kinetic parameters were converted into energy barriers using Eq. (5):

$$\Delta G^{\ddagger}_i = -2.3RT \log(k) \ or \ k = 10^{-\frac{\Delta G^{\ddagger}_i}{2.3RT}} \tag{5}$$

Differential scanning fluorimetry data were fitted to the standard Boltzmann sigmoidal function where Bottom refers to the lowest value, Top is the largest and V50 is the median of those two values:

$$y = Bottom + \frac{(Top - Bottom)}{1 + e^{\frac{V50 - x}{Slope}}} \tag{6}$$

Pre-steady state multiple turnover data were fitted to a standard exponential equation followed by a linear slope:

$$y = (Y_0 - Plateau)e^{-kt} + ax + b \tag{7}$$

For Eq. (7), t is time in seconds, $Y_0$ is the lowest value of Y, Plateau is the Y value in which the exponential phase ends, $k$ is the rate constant in $s^{-1}$, a is the slope of the linear phase and b is the y intercept from the linear phase.

For data fitting using Kintek Global Explorer[56], the following Model was used, in which CDP is either cFP or cWS:

Eox + CDP = Eox.CDP
Eox.CDP = Ered.oxCDP
Ered.oxCDP = Eox + oxCDP

The signal observed was set as below, where scale_1a is a concentration scaling factor, a is the fluorescence contribution for the species being detected, Ered.oxCDP is the NdasCDO-oxidized product complex and oxCDP is the oxidized product, b is the background:

scale_1a*(a*(Ered.oxCDP+oxCDP)+b)

### Reporting summary
Further information on research design is available in the Nature Portfolio Reporting Summary linked to this article.

### Data availability
Structure for Ndas1146 (Uniprot: D7B1W6) and Ndas1147 (Uniprot: D7B1W7) are available with accession codes 9EXV (Ndas1146), and EMD-50049 (Ndas1147). Raw small molecule mass spectrometry data is available on the Figshare project "Broad substrate scope C-C oxidation in cyclodipeptides catalysed by a flavin-dependent filament" (https://doi.org/10.6084/m9.figshare.25556850). The mass spectrometry proteomics data demonstrating covalent attachment of FMN cofactor have been deposited to the ProteomeXchange Consortium via the PRIDE partner repository with the dataset identifier PXD058613. The EPR spectroscopic data supporting this publication can be accessed at https://doi.org/10.17630/2e946d01-0753-45ad-aaf3-87572eef9ec0. Data generated in this study are provided in the Supplementary Information/Source Data file. The Supplementary Information file includes supplementary methods, Supplementary Figs., tables, and supplementary note 1. Supplementary Table 4 contains information about the cryo-EM structure reported here. Source data are provided with this paper.

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

## Acknowledgements

We thank the University of Dundee Cryo-EM facility for access to the instrumentation, funded by Wellcome (223816/Z/21/Z) and MRC (MRC World Class Laboratories PO 4050845509). We thank electron Bio-Imaging Centre (eBIC) facility, Diamond light source Ltd, UK for collection of 300 kV electron microscope data. E.S. was funded by the Cunningham Trust (PhD-CT-18-41), C.M.C. is funded by the Wellcome Trust (210486/Z/18/Z and [204821/Z/16/Z] to the University of St Andrews), B.E.B. acknowledges equipment funding by BBSRC (BB/R013780/1), R.S. is funded by the Wellcome Trust (223816/Z/21/Z).

## Author contributions

Manuscript was written by E.S. in its majority, but all authors contributed to the final form. All authors have given approval to the final version of the manuscript. Specific contributions are as follows: E.S. designed and performed experiments, interpreted data, wrote manuscript; C.J.H. contributed to protein cryo-EM experiments, interpreted data, revised manuscript; T.M.D.M.D.B synthesized cWS and contributed to activity assays, analysed data, revised manuscript; G.J.F. supervised T.M.D.M.D.B, analysed data, revised manuscript; K.A. performed EPR experiments, analysed data, revised manuscript; S.S. performed protein mass spectrometry experiments, analysed data, revised manuscript; B.E.B. designed and performed EPR experiments, analysed data, revised manuscript; R.S. prepared cryo-EM grids, acquired data, analysed data, revised manuscript; C.M.C. participated in project conception, analysed and interpreted data, revised manuscript.

## Competing interests

The authors declare no competing interests.
