## [Transparent Peer Review file · Nature Communications]

Broad substrate scope C-C oxidation in cyclodipeptides catalysed by a flavin-dependent filament

Corresponding Author: Dr Clarissa Czekster

Version 0:

Reviewer comments:

Reviewer #1

(Remarks to the Author)

This manuscript represents a large body of work on the structural and kinetic characterization of a model cyclodipeptide oxidase (CDO), named NdasCDO, which participates in nocazine assembly by *Nocardioopsis dassonvillei*.

A serious concern about this work is the relatively minimal coverage provided of similar prior work on model CDO AlbAB. Andreas and Giessen (ref 41 from BioRxiv in manuscript and published in Nat Comm in April 2024) recently reported the structural characterization of AlbAB, a filamentous CDO homologous to the one characterized herein and reached similar conclusions. It is surprising that this prior highly analogous work is not even cited in the literature review of the Introduction section of the manuscript. While this prior work is briefly mentioned a couple of times within the Results & Discussion, the authors must more fully disclose how their CDO findings compare and contrast with prior findings. They need to present their work in the full context of this previous work in order for the significance of their findings to be more fully understood and assessed.

Minor Concerns:

-To enable the work to be replicated, please provide the codon-optimized DNA sequences of the cloned genes rather than just their protein products. If the actinomycete DNA sequence was not codon optimized, this should be noted along with accession # for the actinomycete gene.

-The information in Table 1 is redundant of Figure 4B. For conciseness of the main text, it is recommended to move the Table in to the Supplemental Information.

Reviewer #2

(Remarks to the Author)

The manuscript by Emmajay Sutherland et. al. presents a comprehensive investigation into the properties and functionalities of NdasCDO, an essential enzyme involved in the nocazine biosynthetic pathway of *Nocardioopsis dassonvillei*. Through a multifaceted approach integrating various biochemical and biophysical techniques, including cryo-electron microscopy (cryo-EM), electron paramagnetic resonance (EPR), and a range of kinetic assays they delve into the workings of NdasCDO. Their exploration extends to elucidating the enzyme's substrate scope and unveiling the critical steps governing the reaction kinetics.

The cryoEM analysis presented provides crucial insights into the structural organization and functional attributes of NdasCDO filaments. Through helical reconstruction, a 3.0 Å resolution map was generated, revealing a filament structure consisting of alternating repeating units of homodimers of subunit A and B forming a heterotetrametric arrangement. The authors have conducted helical reconstruction using the single-particle analysis feature in CryoSPARC, combined with a local refinement strategy. Their careful methodological approach demonstrates a technically rigorous examination of the structure. The resulting characterization reveals a comprehensive and detailed comprehension of the structural complexities. The experimental approach is sound and solid, data analysed rigorously, and the paper is well written. It is worth highlighting and commending the authors' meticulousness and rigor in describing the methods employed and providing detailed explanations of them. Additionally, the quality of the figures and the supplementary information provided with the

study is noteworthy and appreciated. However, some changes are required before publishing:

Major points.

While the manuscript provides a detailed and rigorous account of the experimental results, it falls short in terms of meaningful discussion and contextualization within the field. The mixed Result and Discussion section predominantly focuses on presenting the findings without delving into their broader implications or connecting them to existing literature. This lack of substantial discussion detracts from the overall impact of the. The authors should enrich the manuscript by incorporating more extensive discussions within each section, providing critical analysis of the results, and offering insights into their significance in relation to prior research. By adding thoughtful discussions, the authors can enhance the depth and relevance of their findings, fostering a more robust understanding of the topic among readers and contributing meaningfully to the scientific discourse.

Minor points

1. FMN cofactor (Flavin Mononucleotide cofactor) abbreviation, used for the first time in the abstract, have to be defined as it is not obvious for the non-expert reader.

2. EPR abbreviation is used for the first time in pp 4 ll 13 but not defined until pp7 ll 8.

3. pp6 ll8. The authors state "Grids of the NDS protein were then prepared by applying 4 μ L of 4 μ M protein to a glow-discharged (1 min at 0.1 atmospheric pressure, 35 mAmps, Quorum technologies SC7620) R2/1 400 mesh Cu/Rh holey carbon grids".

In general, the pressure for glow discharge treatments is often in the range of 0.1 mBar (~0.0001 atmospheric pressure).

Glow discharge at 0.1 atmospheric pressure seems a typo. Please, clarify.

Does "R2/1 400 mesh Cu/Rh holey carbon grids" mean Quantifoil R2/1 400 mesh Cu/Rh holey carbon grids?

4. pp 6 ll 16. "...filaments are then transported..." to "...filaments were then transported..."

5. pp 6 ll 20. 0.831 \AA /pixel is original super resolution pixel size or after binning. If the later (expected) I should be specified.

6. pp 6 ll 21. Although total electron dose is specified in Table S3, for consistency should be specified with the exposure time in this line.

7. I would suggest consistency using thousand separator comma (sometimes is used, sometimes not). Consistent uses along the text would improve readability, especially in Materials and Methods section.

8. pp12 ll20-21 reference to Figures S4 and S2 seems swapped.

9. pp 12 ll 34. Some discussion about the interpretation of this experimental MW (i.e. in term of subunits at this pH) would be expected.

10. pp 12 ll 16. "A and b" to "A and B".

11. pp 12 ll 34. It seems the wrong figure panel is mentioned here: "...that higher salt concentrations were associated with higher melting temperatures (Figure 1a).".

12. Same error seems to happen here: "NdasCDO depicted bell shaped profiles for both kcat and kcat/KM (Figure 1b). Best fitted"

13. pp 23 ll 17. There is a repetition in this sentence: "To confirm H2O2 formation in the NdasCDO-catalyzed reaction we used a used a coupled assay".

14. pp23 ll22. The relevant figure has not been annotated in this paragraph: "Burst of product formation...".

15. pp 27 ll15. What does "crystallography experiments" mean in the context of this manuscript?

16. In figure S8, please make sure that the box size is 340 \AA not 340 px.

17. It would be useful adding the information about how you found the helical parameters. Did you get them from a previous study, or you ran a helical parameter search? If you used helical parameter search, it would be informative to include the data from that step and add it to the pipeline.

18. In figure 2, to keep the figure direction consistent, it would be better to replace the position of panel c with panel d. Also, it would be nice to increase the font size of panel C for better demonstration.

Reviewer #3

(Remarks to the Author)

Reviewer #5

(Remarks to the Author)

The manuscript "Broad substrate scope C-C oxidation in cyclodipeptides catalysed by a flavin-dependent filament" by Sutherland et al. describes the detailed characterization of the enzyme NdasCDO. This enzyme oxidizes cyclic dipeptides and could be used in a variety of biotechnology applications, not least because cyclic dipeptides are valuable as potential therapeutics. The authors test the enzyme's stability, its ability to turn over different substrates and determine the cryo-EM structure. Experimentally, the authors do a commendable job to be as thorough as possible and provide very high-quality data. The enzyme and its application will be of interest to the readership of Nature Communications. In its current form though the manuscript is not suitable for the general readership of the journal.

In addition to a plethora of typos, the manuscript is verbose and at times is very difficult to follow. I would urge the authors to shorten their manuscript for clarity and reduce the amount of information displayed in the main-text figures. This would allow the key findings to take center stage. The characterisation for example could be condensed to a short paragraph and most of Figure 1 be moved to the SI.

It would be very helpful for the uninitiated reader to have a proper introductory figure 1.

What is currently missing is a link between the structural data and the enzymatic activity. The authors could improve the paper significantly by modelling the substrates in their cryo-EM structure (simple place-minimize would do) as this would also clearly point to mutations others could use to alter the substrate spectrum of the enzyme for other applications. I am looking forward to reading a suitably revised manuscript.

Reviewer #6

(Remarks to the Author)

In "Broad substrate scope C-C oxidation in cyclodipeptides catalysed by a flavin-dependent filament", Sutherland et al. describe the structure, the molecular and enzymatic capabilities of a CDO enzyme from *Nocardioopsis dassonvillei*. The methods employed seem sound and well performed. They employ a similar approach to that outlined in a previous study (<https://www.nature.com/articles/s41467-024-48030-9>), which they cite the corresponding preprint (reference 41 – line 760). Notably, they are the first to describe the filamentous structure of these enzymes, albeit from a different strain. While the paper is well-executed, its novelty may be limited and the manuscript might be a better fit for a more specialized journal. However, we would leave the decision about novelty up to the editor.

- What are the noteworthy results?

The full structural and kinetic description of a CDO member

- Will the work be of significance to the field and related fields? How does it compare to the established literature? If the work is not original, please provide relevant references.

The novelty of the results are somewhat limited as there seems to be some overlap with another study:
<https://www.nature.com/articles/s41467-024-48030-9>

- Does the work support the conclusions and claims, or is additional evidence needed?

yes

- Are there any flaws in the data analysis, interpretation and conclusions? - Do these prohibit publication or require revision?

Yes, more details are shown afterwards

- Is the methodology sound? Does the work meet the expected standards in your field?

Yes

- Is there enough detail provided in the methods for the work to be reproduced?

In part

Detailed Comments:

The methodology of the substrate evaluation through LC-MS is not well described. The figures (especially supplementary ones) explanation could be improved to allow a better verification.

Overall we would suggest the authors to carefully check the correspondence between Figure number (in particular with supplementary figures) and actual number, which in some cases is wrong.

Line 636: "Inset shows in blue a unique transition if cFP-2H underwent dehydrogenation on the Pro sidechain, while purple (not present) depicts unique transition if cFP-2H underwent dehydrogenation on the Phe sidechain. Fragmentation was determined as previously described" – There must be an error cause the figure 4C is not about cFP but Clp:

Line 86: Intriguingly, efforts have successfully combined CDPS and CDOs from different organisms within a singular vector, yielding a diverse array of dehydrogenated CDPs²². Nevertheless, a deeper molecular understanding of CDO structure and mechanism promises to enhance their utility as biocatalysts.

Line 412 Figure S4 This supplementary Figure is not NdasCDO absorption spectrum

Line 413 Figure S2 Wrong Supplementary Figure, it should be S4

-Cys121: Please explicit the shifted mass values for the Y27 fragment compatible with FMN binding. To do that I suggest to also perform a digestion of the enzyme itself without FMN as a control.

SUPPLEMENTARY

Page 17, Figure S4c: They don't say which MS difference the y27 Cys 121 fragment presents in order for them to understand this is binding FMN

Pages 11 and 12, S3c and S3d present a partial mass spectrum that makes impossible its evaluation.

Figure S3 a-h captions is absent and are not really explanatory. Which is the control, what does mean peptides no CDO, and CDO not peptides? Please assign letters to each spectrum and explain it to allow a good evaluation.

Reviewer #7

(Remarks to the Author)

I co-reviewed this manuscript with one of the reviewers who provided the listed reports. This is part of the Nature Communications initiative to facilitate training in peer review and to provide appropriate recognition for Early Career

Researchers who co-review manuscripts

Version 1:

Reviewer comments:

Reviewer #2

(Remarks to the Author)

The authors have addressed all my comments and concerns. In my opinion, the reviewed manuscript is now suitable for publication in its current form.

Reviewer #5

(Remarks to the Author)

I thank the authors for addressing all points raised by me to my full satisfaction and recommend publication without further revision.

Reviewer #6

(Remarks to the Author)

Thanks a lot for addressing our suggestions and pointing out the relevance in comparison to the other recent paper. As mentioned before we would leave the question about novelty up to the editor, and all other points have been address to our satisfaction.

Reviewer #7

(Remarks to the Author)

We thank the editor and reviewers for careful consideration of our manuscript. We present a point-by-point response to the points raised in **blue**. Text modified in the manuscript or supporting information is shown in **orange**.

Reviewer's Comments:

Reviewer #1 (Remarks to the Author):

This manuscript represents a large body of work on the structural and kinetic characterization of a model cyclodipeptide oxidase (CDO), named NdasCDO, which participates in nocazine assembly by *Nocardiosis dassonvillei*.

We thank the reviewer for careful consideration of our manuscript.

A serious concern about this work is the relatively minimal coverage provided of similar prior work on model CDO AlbAB. Andreas and Giessen (ref 41 from BioRxiv in manuscript and published in Nat Comm in April 2024) recently reported the structural characterization of AlbAB, a filamentous CDO homologous to the one characterized herein and reached similar conclusions. It is surprising that this prior highly analogous work is not even cited in the literature review of the Introduction section of the manuscript. While this prior work is briefly mentioned a couple of times within the Results & Discussion, the authors must more fully disclose how their CDO findings compare and contrast with prior findings. They need to present their work in the full context of this previous work in order for the significance of their findings to be more fully understood and assessed.

We thank the reviewer for highlighting this. The now published manuscript from Andreas and Giessen had not been accepted for publication when we first submitted this manuscript. We were in contact with the editorial office who were fully aware both manuscripts would be under review at the time of our initial submission.

Although both manuscripts detail the structure of a filament enzyme, our work thoroughly characterizes the reaction, proposes intermediates and structural features underlying catalysis. At the time of our original submission in April we did not have access to a peer-reviewed manuscript from Andreas and Giessen, or to files with structural data for accurate comparison. We are pleased that now this is available, and we have fully placed our results into context as highlighted below. We also added Figure S8 with a direct comparison of the structures.

Introduction states:

“Recently, it was determined that a member of the CDO family formed a heterotetrameric filamentous structure from the two individual subunit dimers.²¹ Enzymatic activity was highly dependent on filament formation which necessitated the simultaneous expression of both subunits; attempts at individual subunit purification were also unsuccessful by Giessen et al. While enzyme filaments have been previously described,²² structural and mechanistic characterisations of these proteins are difficult, thus limiting a comprehensive understanding of reaction mechanism and structural data.”

In the Results and discussion we say:

“Comparison to the only known other CDO structure, AlbAB, highlighted a high degree of similarity between each A subunit (Figure S8a) with an RMSD of 0.809 Å across 158

pruned atom pairs (as calculated by MatchMaker in UCSF ChimeraX, version 1.7.1). This is not unexpected given that these enzymes are from the same family and further supports the work by Giessen *et al.*”

Minor Concerns:

-To enable the work to be replicated, please provide the codon-optimized DNA sequences of the cloned genes rather than just their protein products. If the actinomycete DNA sequence was not codon optimized, this should be noted along with accession # for the actinomycete gene.

We thank the reviewers for bringing this to our attention and have included the DNA sequences of Ndas1146 and Ndas1147 to the SI.

-The information in Table 1 is redundant of Figure 4B. For conciseness of the main text, it is recommended to move the Table in to the Supplemental Information.

Table 1 has now been moved to SI under the heading Table S3.

Reviewer #2 (Remarks to the Author)

The manuscript by Emmajay Sutherland *et al.* presents a comprehensive investigation into the properties and functionalities of NdasCDO, an essential enzyme involved in the nocazine biosynthetic pathway of *Nocardioopsis dassonvillei*. Through a multifaceted approach integrating various biochemical and biophysical techniques, including cryo-electron microscopy (cryo-EM), electron paramagnetic resonance (EPR), and a range of kinetic assays they delve into the workings of NdasCDO. Their exploration extends to elucidating the enzyme's substrate scope and unveiling the critical steps governing the reaction kinetics.

The cryoEM analysis presented provides crucial insights into the structural organization and functional attributes of NdasCDO filaments. Through helical reconstruction, a 3.0 Å resolution map was generated, revealing a filament structure consisting of alternating repeating units of homodimers of subunit A and B forming a heterotetrametric arrangement. The authors have conducted helical reconstruction using the single-particle analysis feature in CryoSPARC, combined with a local refinement strategy. Their careful methodological approach demonstrates a technically rigorous examination of the structure. The resulting characterization reveals a comprehensive and detailed comprehension of the structural complexities.

The experimental approach is sound and solid, data analysed rigorously, and the paper is well written. It is worth highlighting and commending the authors' meticulousness and rigor in describing the methods employed and providing detailed explanations of them. Additionally, the quality of the figures and the supplementary information provided with the study is noteworthy and appreciated. However, some changes are required before publishing:

We thank the reviewer for their time and effort reviewing our manuscript. We are also grateful for the supporting comments on the work and manuscript.

Major points.

While the manuscript provides a detailed and rigorous account of the experimental results, it falls short in terms of meaningful discussion and contextualization within the field. The mixed Result and Discussion section predominantly focuses on presenting the findings without delving into their broader implications or connecting them to existing literature. This lack of substantial discussion detracts from the overall impact of the. The authors should enrich the manuscript by incorporating more extensive discussions within each section, providing critical analysis of the results, and offering insights into their significance in relation to prior research. By adding thoughtful discussions, the authors can enhance the depth and relevance of their findings, fostering a more robust understanding of the topic among readers and contributing meaningfully to the scientific discourse.

We take this comment on board and have therefore further added instances where results are put into perspective. One slight challenge we encountered is that this is the first FMN-dependent enzyme catalysing peptide dehydration without the need for additional reductants/oxidizers (such as NADH for example) for which such kinetic characterization has been carried out, a system made further unique by the enzyme acting as a filament. This is a challenge and an opportunity. Nevertheless, similar experiments carried out with other flavin dependent enzymes were incorporated into additional “results and discussion”. Since another reviewer thought we needed to shrink the manuscript we strived to strike a balance between adding more discussion while not deviating focus on our findings.

Some of these instances are shown below:

“Covalently attached flavin cofactors are observed in approximately 10% of flavoenzymes, and this attachment has been proposed to modulate cofactor redox potential, therefore facilitating catalysis of thermodynamically challenging reactions.³⁹⁴⁰ Additionally, the reaction catalysed by NdasCDO bears similarity to internal monooxygenases, as the same substrate is oxygenated (or dehydrated in the case of NdasCDO) by the enzyme and also subsequently acts as electron donor in flavin reduction (Figure 2a).⁴¹”

“Modelling CDP substrates into a substrate binding pocket in the vicinity of the FMN cofactor suggests that substrates such as cFG sample fewer conformations than substrates that can undergo two oxidation events (cFP, cWF, cWS, cWY, cLP, cHF), and more conformations are sampled by substrates that contain tryptophan, in agreement with their lower k_{cat}/K_M values (Figure S12).”

Minor points

1. FMN cofactor (Flavin Mononucleotide cofactor) abbreviation, used for the first time in the abstract, have to be defined as it is not obvious for the non-expert reader.

We thank the reviewer for this point and apologise for any misunderstanding this may have caused. The manuscript has been updated accordingly: “We show NdasCDO forms filaments in solution with a covalently bound flavin mononucleotide (FMN) cofactor in the interface between three distinct subunits.”

2. EPR abbreviation is used for the first time in pp 4 ll 13 but not defined until pp7 ll 8.

We thank the reviewer for highlighting this discrepancy and have updated the

manuscript accordingly: “...electron paramagnetic resonance (EPR), ...” on page 5, line 114.

3. pp6 ll8. The authors state “Grids of the NDS protein were then prepared by applying 4 μ L of 4 μ M protein to a glow-discharged (1 min at 0.1 atmospheric pressure, 35 mAmps, Quorum technologies SC7620) R2/1 400 mesh Cu/Rh holey carbon grids”.

In general, the pressure for glow discharge treatments is often in the range of 0.1 mBar (~0.0001 atmospheric pressure). Glow discharge at 0.1 atmospheric pressure seems a typo. Please, clarify.

We thank the reviewer for highlighting this mistake and have since rectified it to 0.1 mbar.

Does “R2/1 400 mesh Cu/Rh holey carbon grids” mean Quantifoil R2/1 400 mesh Cu/Rh holey carbon grids?

We apologise for the miscommunication and have since clarified the statement to include ‘Quantifoil’.

4. pp 6 ll 16. “...filaments are then transported...” to “...filaments were then transported...”

We thank the reviewers for this comment and have since updated the manuscript.

5. pp 6 ll 20. 0.831 \AA /pixel is original super resolution pixel size or after binning. If the later (expected) I should be specified.

We apologise for not previously specifying this but the manuscript now includes: “0.831 \AA /pixel after 2X binning”.

6. pp 6 ll 21. Although total electron dose is specified in Table S3, for consistency should be specified with the exposure time in this line.

We apologise for this inconsistency but have updated the Methods section to explicitly state:

“In total, 12900 movies were collected with 50 frames per movie stack with a total dose of 33.44 $e/\text{\AA}^2$ and exposure time of 1.79 seconds per movie stack using aberration free image shift (AFIS).”

7. I would suggest consistency using thousand separator comma (sometimes is used, sometimes not). Consistent uses along the text would improve readability, especially in Materials and Methods section.

We apologise for any inconsistencies and inconveniences this may have caused. All numbers have been modified to remove the thousand separator comma throughout the manuscript.

8. pp12 ll20-21 reference to Figures S4 and S2 seems swapped.

We thank the reviewer for highlighting this point which has since been corrected through the manuscript.

9. pp 12 ll 34. Some discussion about the interpretation of this experimental MW (i.e. in term of subunits at this pH) would be expected.

We thought this fitted better with the discussion about oligomeric state and therefore on pp12 we state “also discussed further below under “*Oligomeric state*””

There we state:

“Functional enzyme contains subunit A (monomer MW = 22 kDa) and subunit B (monomer MW = 11 kDa) at an unknown ratio.”

10. pp 12 ll 16. “A and b” to “A and B”.

We can't identify this instance unfortunately but have checked the nomenclature for subunits A and B assuming it was referring to that, correcting instances in which it was not correct.

11. pp 12 ll 34. It seems the wrong figure panel is mentioned here: “...that higher salt concentrations were associated with higher melting temperatures (Figure 1a).”.

We changed this to state “Figure 1c”

12. Same error seems to happen here: “NdasCDO depicted bell shaped profiles for both kcat and kcat/KM (Figure 1b). Best fitted”

We changed this to state “Figure 1d”

13. pp 23 ll 17. There is a repetition in this sentence: “To confirm H₂O₂ formation in the NdasCDO-catalyzed reaction we used a used a coupled assay”.

We rephrased to correct:

To confirm H₂O₂ formation in the *NdasCDO*-catalyzed reaction we used a coupled assay with hydrogen peroxidase and ABTS (2,2'-Azinobis [3-ethylbenzothiazoline-6-sulfonic acid]-diammonium salt). An increase in absorbance at 560nm only in the presence of *NdasCDO* and cFP, consistent with H₂O₂ formation, was observed

14. pp23 ll22. The relevant figure has not been annotated in this paragraph: “Burst of product formation...”.

There should be a reference to Figure 6b and 6c, we added it where suggested.

15. pp 27 ll15. What does “crystallography experiments” mean in the context of this manuscript?

We apologize for this, although C.J.H. did attempt to crystallize the complex filament structure these were unsuccessful. He then participated in setting up grids for cryo-EM, and participation was amended to reflect that.

16. In figure S8, please make sure that the box size is 340 Å not 340 px.

Box size is referred to as 340 Å.

17. It would be useful adding the information about how you found the helical parameters. Did you get them from a previous study, or you ran a helical parameter search? If you used helical parameter search, it would be informative to include the data from that step and add it to the pipeline.

We thank the reviewer for highlighting this need for more information regarding the helical parameters which has now been included in the methods section:

“Helical parameters including helical rise and twist were searched within the reconstructed volume and the initial estimates were input for the Helix refine job to yield a final estimated helical twist of 132.3° and a helical rise of 45.45 \AA .”

18. In figure 2, to keep the figure direction consistent, it would be better to replace the position of panel c with panel d. Also, it would be nice to increase the font size of panel C for better demonstration.

We thank the reviewer for this suggestion and have incorporated these changes into the figure which is now Figure 3 in the manuscript (shown below).

Reviewer #3 (Remarks to the Author)

Reviewer #5 (Remarks to the Author)

The manuscript "Broad substrate scope C-C oxidation in cyclodipeptides catalysed by a flavin-dependent filament" by Sutherland et al. describes the detailed characterization

of the enzyme NdasCDO. This enzyme oxidizes cyclic dipeptides and could be used in a variety of biotechnology applications, not least because cyclic dipeptides are valuable as potential therapeutics. The authors test the enzyme's stability, its ability to turn over different substrates and determine the cryo-EM structure. Experimentally, the authors do a commendable job to be as thorough as possible and provide very high-quality data. The enzyme and its application will be of interest to the readership of Nature Communications. In its current form though the manuscript is not suitable for the general readership of the journal.

We thank the reviewer for their time and effort reviewing our manuscript, and have addressed the points raised below.

In addition to a plethora of typos, the manuscript is verbose and at times is very difficult to follow.

We apologize and have fixed any spelling errors we encountered upon review. Without knowing which instances are considered verbose we made some modifications, but ultimately think that because the paper contains substantial kinetic assays combined with an unusual structure of an enzyme participating in natural product biosynthesis the manuscript is likely to reach a broad readership, and therefore have made every effort possible to make it readable and interpretable by different audiences.

I would urge the authors to shorten their manuscript for clarity and reduce the amount of information displayed in the main-text figures. This would allow the key findings to take center stage.

We thank the reviewer for this comment. Reviewer #2 suggested we add further points for discussion, so although the manuscript has not been shortened per se, we added focus on the main findings and objectives. We want to emphasize that a similar chemical reaction has not been thoroughly characterized for a flavoenzyme or a filament enzyme. We therefore think it is crucial to state findings and rationale for carrying out experiments.

The characterisation for example could be condensed to a short paragraph and most of Figure 1 be moved to the SI.

We thank the reviewer for this comment. As per the suggestion below, we added a first “summary figure”. We made changes moving some elements of Figure 1 to this summary, and others to SI. However, it is our view that it is crucial to display raw experimental data as well a general scheme for the reaction with different substrates since not having those in the main paper makes interpreting kinetic data very difficult. We therefore kept some of the raw data on the characterisation, specifically those pertaining to mechanistic discussions later on.

It would be very helpful for the uninitiated reader to have a proper introductory figure 1. We agree with this point and have now included an introductory figure.

What is currently missing is a link between the structural data and the enzymatic activity.

We thank the reviewer for this comment. Now the final section of the manuscript “Mechanism, structure and catalytic turnover by a filament enzyme” brings together the

main findings and power of combining structure with kinetics. Specifically, we discuss results from mutants (N-terminal truncations and S58A mutation), as well as residues that are likely participating in catalysis and substrate selection.

Furthermore, as suggested below we carried out docking simulations with the cyclic dipeptide substrates we tested as substrates, further linking structural and kinetic data.

To make clearer how these are intertwined we have made changes:

“Modelling CDP substrates into a substrate binding pocket in the vicinity of the FMN cofactor suggests that substrates such as cFG sample fewer conformations than substrates that can undergo two oxidation events (cFP, cWF, cWS, cWY, cLP, cHF), and more conformations are sampled by substrates that contain tryptophan, in agreement with their lower k_{cat}/K_M values (Figure S12).”

“Whilst *NdasCDO-A* harbours high structural similarity to nitroreductases, our work highlights that the active site of CDOs differs from these enzymes, as it does not rely on the same catalytic mechanism and structural features. Instead, *NdasCDO* is a unique flavoenzyme⁴¹, operating optimally in high salt and high pH conditions without the need for additional cofactors, in a reaction mechanism limited not by the complex three-dimensional arrangement of the protein, but by cofactor regeneration. Combining our kinetic and structural data we put forward a chemical mechanism involving a neutral flavin semiquinone radical intermediate, in an initial fast phase in which Ser58-A and Tyr36-B play a crucial role.”

The authors could improve the paper significantly by modelling the substrates in their cryo-EM structure (simple place-minimize would do) as this would also clearly point to mutations others could use to alter the substrate spectrum of the enzyme for other applications.

We thank the reviewer for this suggestion. We have modelled several substrates under study (cFG, cFP, cHF, cLP, cWF, cWS and cYW) and added the methods (in the supporting information) and embedded results into our substrate selection discussion.

I am looking forward to reading a suitably revised manuscript.

We thank the reviewer for their time and effort and hope this version of the manuscript is satisfactory.

Reviewer #6 (Remarks to the Author):

In "Broad substrate scope C-C oxidation in cyclodipeptides catalysed by a flavin-dependent filament", Sutherland et al. describe the structure, the molecular and enzymatic capabilities of a CDO enzyme from *Nocardioopsis dassonvillei*. The methods employed seem sound and well performed. They employ a similar approach to that outlined in a previous study (<https://www.nature.com/articles/s41467-024-48030-9>), which they cite the corresponding preprint (reference 41 – line 760). Notably, they are the first to describe the filamentous structure of these enzymes, albeit from a different strain. While the paper is well-executed, its novelty may be limited and the manuscript

might be a better fit for a more specialized journal. However, we would leave the decision about novelty up to the editor.

We thank the reviewer for the time and effort reviewing our manuscript. At the time of submission both manuscripts were under review at Nat Comms, and the editor was aware of both manuscripts.

Research by Andreas and Giessen is noteworthy for being the first published structure of a cyclodipeptide oxidase. Our work is comprehensive and highlights that CDOs are highly promiscuous and can oxidise substrates outside of their biosynthetic pathways, with a clear proposal for catalytic and chemical mechanisms. Additionally, although there is a plethora of studies on mechanisms and catalysis by flavoenzymes, this is the first report we are aware of describing the chemical and catalytic mechanism of a flavin-dependent filament enzyme. Furthermore, CDOs catalyse their reaction without the need for additional substrates or cofactors, and therefore understanding these enzymes holds great potential for enzyme design and utilization in biocatalysis.

- What are the noteworthy results?

The full structural and kinetic description of a CDO member

- Will the work be of significance to the field and related fields? How does it compare to the established literature? If the work is not original, please provide relevant references. The novelty of the results are somewhat limited as there seems to be a some overlap with another study: <https://www.nature.com/articles/s41467-024-48030-9>

As we mentioned above, this is the first report describing the chemical and catalytic mechanism of a flavin-dependent filament enzyme, and our results expand significantly beyond structure.

- Does the work support the conclusions and claims, or is additional evidence needed?
yes

- Are there any flaws in the data analysis, interpretation and conclusions? - Do these prohibit publication or require revision?

Yes, more details are shown afterwards

- Is the methodology sound? Does the work meet the expected standards in your field?
Yes

- Is there enough detail provided in the methods for the work to be reproduced?
In part

Detailed Comments:

The, the methodology of the substrate evaluation through LC-MS is not well described. On the main paper materials and methods, on the section “**LC-MS Analysis of oxidised cyclodipeptides**”, we fully describe the LC-MS methods employed, both in terms of chromatography and mass spectrometry data acquisition. We thank the reviewer for this observation though, as the time courses analysed by mass spectrometry to

determine oxidation order could have been better described. We therefore added specific methods for the time courses as follows:

Time course reactions

The reaction stock was prepared in 50 mM Tris pH 9 and 200 mM NaCl with the CDP substrate (cFP or cLP) at 25 μ M. *NdasCDO* was added last to a final concentration of 0.1 μ M and the reaction was incubated at 30 °C. At each time point, 40 μ L were removed and boiled at 100 °C for 5 mins then centrifuged to pellet the denatured protein. The supernatant was then analysed via LC-MS as described previously.

The figures (especially supplementary ones) explanation could be improved to allow a better verification.

We believe this could be in reference to Figure S3, we added this to the legend:

-2H denotes products with a single oxidation, and -4H denotes substrates that underwent two oxidation events.

Overall we expanded Figure legends in the SI as much as possible.

Overall we would suggest the authors to carefully check the correspondence between Figure number (in particular with supplementary figures) and actual number, which in some cases is wrong.

We apologize for this, all references to figures have been reviewed and confirmed to be correct.

Line 636: “Inset shows in blue a unique transition if cFP-2H underwent dehydrogenation on the Pro sidechain, while purple (not present) depicts unique transition if cFP-2H underwent dehydrogenation on the Phe sidechain. Fragmentation was determined as previously described” – There must be an error cause the figure 4C is not about cFP but Clp:

The reviewer is correct, and we apologize for this. The panel on 4d should depict the oxidation of cLP to accompany 4c. We have fixed this, as well as the Figure legend and moved cFP oxidation to Figure S14.

Line 86: Intriguingly, efforts have successfully combined CDPS and CDOs from different organisms within a singular vector, yielding a diverse array of dehydrogenated CDPS²². Nevertheless, a deeper molecular understanding of CDO structure and mechanism promises to enhance their utility as biocatalysts.

We are not sure what the comment is but believe it is due to a typo (CDPS²²). We have fixed this instance to reflect.

Line 412 Figure S4 This supplementary Figure is not *NdasCDO* absorption spectrum

Line 413 Figure S2 Wrong Supplementary Figure, it should be S4

We apologise for any discrepancies in the figure references highlighted here and have since rectified these throughout the manuscript.

-Cys121: Please explicit the shifted mass values for the Y27 fragment compatible with FMN binding. To do that I suggest to also perform a digestion of the enzyme itself without FMN as a control.

We have added the following figure to Figure S5, as well as mass values for expected and observed species to demonstrate covalent attachment of one FMN cofactor/monomer (covalent site demonstrated by trypsin digestion as shown on Figure S5c, mass shifted by 453 Da).

Purifying the complex without FMN is not possible, given that the CDO is only soluble when CDO-A, CDO-B and the FMN subunit are present. Previous work also agrees that subunit A has never been purified alone, therefore trypsin digestion without FMN is not possible.

SUPPLEMENTARY

Page 17, Figure S4c: They don't say which MS difference the y27 Cys 121 fragment presents in order for them to understand this is binding FMN

This is relative to Figure S5c, now reflects the mass shift.

Pages 11 and 12, S3c and S3d present a partial mass spectrum that makes impossible its evaluation. Figure S3 a-h captions is absent and are not really explanatory. Which is the control, what does mean peptides no CDO, and CDO not peptides? Please assign letters to each spectrum and explain it to allow a good evaluation.

We have changed the figure legend and labelling to make interpretation clearer. The figure legend states grey as a control without enzyme, and -2H and -4H as products with one and two oxidations respectively. Figure labelling was expanded to make interpretation easier and independent of the legend.

Reviewer #7 (Remarks to the Author):

I co-reviewed this manuscript with one of the reviewers who provided the listed reports. This is part of the Nature Communications initiative to facilitate training in peer review and to provide appropriate recognition for Early Career Researchers who co-review manuscripts